# Neurexin–Neuroligin 1 regulates synaptic morphology and functions via the WAVE regulatory complex in *Drosophila* neuromuscular junction

Guanglin Xing[1†], Moyi Li[1,2,3†], Yichen Sun[1], Menglong Rui[1], Yan Zhuang[1], Huihui Lv[2], Junhai Han[1,2,3], Zhengping Jia[1,4]*, Wei Xie[1,2,3]*

[1]Institute of Life Sciences, the Collaborative Innovation Center for Brain Science, Southeast University, Nanjing, China; [2]The Key Laboratory of Developmental Genes and Human Disease, Southeast University, Nanjing, China; [3]Co-innovation Center of Neuroregeneration, Nantong University, Nantong, China; [4]Neurosciences and Mental Health Program, The Hospital for Sick Children, University of Toronto, Ontario, Canada

*For correspondence:
zhengping.jia@sickkids.ca (ZJ);
wei.xie@seu.edu.cn (WX)

†These authors contributed equally to this work

Competing interests: The authors declare that no competing interests exist.

**Abstract** Neuroligins are postsynaptic adhesion molecules that are essential for postsynaptic specialization and synaptic function. But the underlying molecular mechanisms of neuroligin functions remain unclear. We found that *Drosophila* Neuroligin 1 (DNlg1) regulates synaptic structure and function through WAVE regulatory complex (WRC)-mediated postsynaptic actin reorganization. The disruption of DNlg1, DNlg2, or their presynaptic partner neurexin (DNrx) led to a dramatic decrease in the amount of F-actin. Further study showed that DNlg1, but not DNlg2 or DNlg3, directly interacts with the WRC via its C-terminal interacting receptor sequence. That interaction is required to recruit WRC to the postsynaptic membrane to promote F-actin assembly. Furthermore, the interaction between DNlg1 and the WRC is essential for DNlg1 to rescue the morphological and electrophysiological defects in *dnlg1* mutants. Our results reveal a novel mechanism by which the DNrx-DNlg1 trans-synaptic interaction coordinates structural and functional properties at the neuromuscular junction.
DOI: https://doi.org/10.7554/eLife.30457.001

## Introduction

Synapses are fundamental components of neural circuits that are essential for normal brain function. The formation and maturation of synapses require coordination between presynaptic and postsynaptic apparatuses and are tightly controlled to ensure proper synaptic structure and function. A number of critical synaptic proteins have been identified (*Kittel et al., 2006*; *Liu et al., 2011*; *Owald et al., 2012*; *Petzoldt and Sigrist, 2014*; *Waites et al., 2005*; *Wu et al., 2010*; *Yamagata et al., 2003*). Among those are the heterotypic synaptic adhesion molecules neurexins and neuroligins, which have captured special attention because of their potent synaptogenic properties and their genetic linkage to autism and other mental disorders (*Feng et al., 2006*; *Jamain et al., 2003*; *Kim et al., 2008*; *Laumonnier et al., 2004*; *Lawson-Yuen et al., 2008*; *Tian et al., 2017*; *Yu et al., 2017*).

Neuroligins are postsynaptic adhesion molecules involved in synaptic formation and maturation and in postsynaptic assembly. An important early discovery regarding neuroligins was that they can induce formation of new synapses *in vitro* (*Scheiffele et al., 2000*). Expression of neuroligins in non-neuronal cells is sufficient to induce presynaptic differentiation in axons that form contacts between

nonneuronal and neuronal cells (*Scheiffele et al., 2000*). The overexpression of neuroligins in neuronal cultures increases the number of spines and induces the accumulation of postsynaptic proteins (*Chih et al., 2005*; *Chubykin et al., 2007*). Because of the existence of multiple, functionally redundant family members, it is difficult to directly assess the effects of neuroligins on synaptic formation *in vivo*. The current data from both mammals and flies strongly support the critical involvement of neuroligins in synaptic function and the maturation of the postsynaptic apparatus (*Banovic et al., 2010*; *Chubykin et al., 2007*; *Poulopoulos et al., 2009*; *Sun et al., 2011*; *Varoqueaux et al., 2006*; *Xing et al., 2014*; *Zhang et al., 2017*). However, it is still not elucidated how neuroligins regulate those processes.

Actin is a key cytoskeletal component that plays a central role in many cellular processes, including cell morphology, motility, and vesicle trafficking (*Kaksonen et al., 2006*; *Le Clainche and Carlier, 2008*). It is well known as a critical factor for synaptic formation and function. In mammalian central synapses, actin cytoskeletons are enriched in the dendritic spines, the major sites of excitatory input, and are crucial for spine formation, morphology, and plasticity (*Dillon and Goda, 2005*; *Takahashi et al., 2003*). Actin dynamics are involved in the postsynaptic cell-surface expression and trafficking of the glutamate receptor (GluR) (*Cingolani and Goda, 2008*) and in various forms of synaptic plasticity, including long-term potentiation and depression, which are widely regarded as synaptic mechanisms for learning and memory (*Collingridge et al., 2010*; *Malenka and Bear, 2004*). Among the many factors that have been characterized to regulate the actin cytoskeleton, the members of the Wiskott–Aldrich syndrome protein (WASP) family have been extensively studied. WASP family proteins are characterized by a conserved C-terminal tripartite verprolin-homology, central, acidic (VCA) domain, which binds to and stimulates the nucleating activity of the Arp2/3 complex to promote actin polymerization (*Pollitt and Insall, 2009*; *Takenawa and Suetsugu, 2007*). The WASP family Verprolin-homologous protein (WAVE) is constitutively incorporated into the WAVE regulatory complex (WRC), a five-component complex that consists of Sra1/Cyfip1, Hem2/Nap1/Kette, Abi2, HSPC300/Brick1, and WAVE/SCAR (*Eden et al., 2002*). Activation of the WASP family members is largely restricted to cell membranes, and several membrane proteins have been shown to bind to WRC to regulate actin dynamics (*Chia et al., 2014*; *Nakao et al., 2008*).

Similar to the mammalian system, *in vivo* studies indicated strong involvement of *Drosophila* neuroligins (DNlgs) in synapse development and function. *Drosophila* has four neuroligin genes (*dnlg1–4*), which have a close evolutionary relationship to their vertebrate homologs (*Banovic et al., 2010*). Using the *Drosophila* neuromuscular junction (NMJ) as a model, we and others previously showed that all four DNlgs play roles in synaptic formation and function, including the regulation of bouton growth, subsynaptic reticulum (SSR) assembly, GluR recruitment, and synaptic transmission (*Banovic et al., 2010*; *Chen et al., 2012*; *Sun et al., 2011*; *Xing et al., 2014*; *Zhang et al., 2017*). Exactly how DNlgs regulate those processes remain to be fully understood. Given the critical role of actin in postsynaptic regulation, we hypothesized that the abnormalities observed in *dnlg*-mutant NMJs are related to actin deficits.

In this study, we reveal a significant role of DNlg1 in the regulation of the actin cytoskeleton in the postsynaptic NMJ. We find that mutations in *dnlg1, dnlg2,* and *dnrx* each resulted in a dramatic reduction in the amount of actin filaments (F-actin). And it is DNlg1, but not DNlg2 or DNlg3, able to directly interact with the WRC via the WRC interacting receptor sequence (WIRS) motif. This motif is present in the C-terminal tail of DNlg1 but is absent in DNlg2 and DNlg3. Mutant DNlg1 that could not bind to WRC failed to reverse NMJ synapse undergrowth and reduced NMJ synaptic transmission capability in *dnlg1* mutants. Altogether, DNlg1 promotes postsynaptic F-actin assembly via binding and recruiting WRC to postsynaptic sites. And that interaction between DNlg1 and WRC is indispensable to maintain normal synaptic formation and transmission in *Drosophila* NMJs. This study unravels a fundamental mechanism how certain synaptic adhesion molecules regulate synaptic formation and function.

## Results

### DNlg1 and DNlg2 positively regulate postsynaptic F-actin assembly

To explore the relationship between neuroligins and the postsynaptic actin cytoskeleton, we analyzed the level of F-actin at the NMJ in *Drosophila* neuroligin mutants. The *Drosophila* body-wall

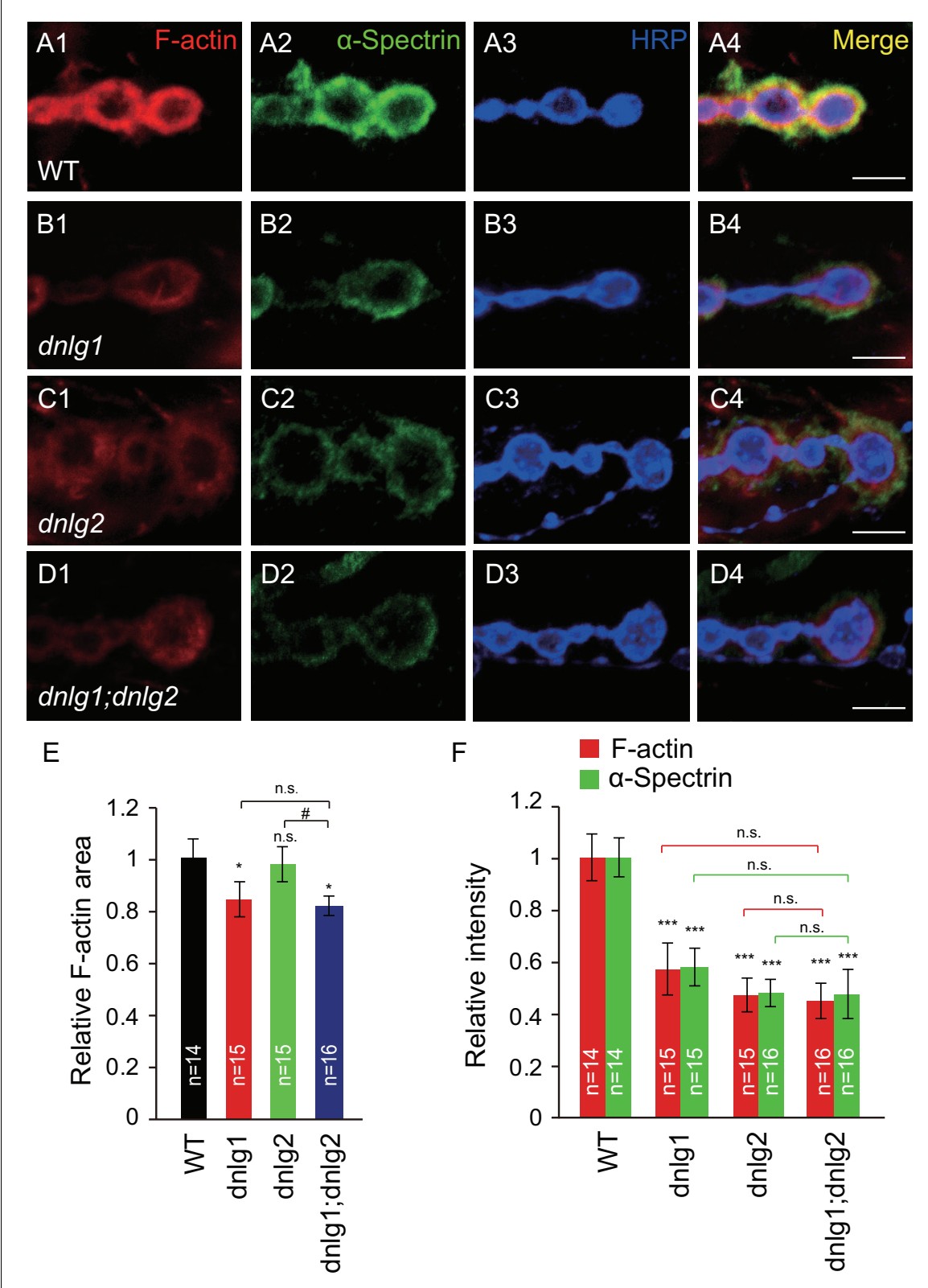

**Figure 1.** DNlg1 and DNlg2 promote postsynaptic F-actin assembly. (A–D) Confocal images of (A) WT, (B) *dnlg1* mutant (*dnlg1*$^{ex1.9/ex2.3}$), (C) *dnlg2* mutant (*dnlg2*$^{KO70/KO70}$), and (D) *dnlg1; dnlg2* double mutant (*dnlg2*$^{KO70/KO70}$; *dnlg1*$^{ex1.9/ex2.3}$) third instar larvae NMJ type Ib boutons at muscles 12/13 labeled with Texas Red phalloidin (red), anti-α-Spectrin (green), and anti-HRP (blue). (E) Summary graph showing a significant decrease in the relative area of F-actin in *dnlg1* single mutants and *dnlg1; dnlg2* double mutants compared with WT. Note that the relative F-actin area is normal in the *dnlg2*

*Figure 1 continued on next page*

*Figure 1 continued*

mutants. (F) Summary graph showing the relative fluorescence intensity of F-actin and α-Spectrin in various genotypes. Note that the relative fluorescence intensity of F-actin and α-Spectrin is significantly decreased in the *dnlg1* single mutants, *dnlg2* single mutants and *dnlg1; dnlg2* double mutants. The data in (E) and (F) are shown as the mean ± SEM; n represents the number of boutons analyzed; asterisks indicate significant differences between WT and the indicated genotypes. *p<0.05; ***p<0.001; n.s., not significant. Hashes indicate significant differences between two indicated groups. #p<0.05; n.s., not significant. Scale bars: (A–D) 5 µm.

DOI: https://doi.org/10.7554/eLife.30457.002

The following source data and figure supplements are available for figure 1:

**Source data 1.** Sample size (n), mean, SEM, and one-way ANOVA (and nonparametric) with Tukey's multiple comparisons test are presented for the data in *Figure 1E and F*.

DOI: https://doi.org/10.7554/eLife.30457.004

**Figure supplement 1.** The distribution and localization of DLG are normal in neuroligin mutants.

DOI: https://doi.org/10.7554/eLife.30457.003

**Figure supplement 1—source data 1.** Sample size (n), mean, SEM, and Mann–Whitney test in *Figure 1—figure supplement 1D*.

DOI: https://doi.org/10.7554/eLife.30457.005

muscles are innervated by numerous motor neurons that branch over the muscles and form stereo-typic NMJ terminal boutons, each containing dozens of glutamatergic synapses (*Prokop, 2006*). F-actin is usually concentrated around the postsynaptic site of the bouton, which can be visualized by using fluorophore-conjugated phalloidin. Consistent with previous studies (*Coyle et al., 2004*), F-actin in the wild type (WT) NMJ was highly enriched at the postsynaptic sites, displaying a diffuse meshwork-like appearance (*Figure 1A*). In the *dnlg1* and *dnlg2* mutant NMJs, the amount of F-actin was dramatically reduced (*Figure 1B,C and F*). The amount of α-Spectrin, an actin cytoskeleton-associated protein, was similarly reduced in the mutants (*Figure 1B,C and F*) *Mosca et al., 2012*. In addition to the lower fluorescence intensity, the *dnlg1* mutants had a significantly less relative area of F-actin distribution [ratio of F-actin area/horseradish peroxidase (HRP)-labeled area; *Figure 1B and E*]. The alterations in F-actin and α-Spectrin were specific, as they were not accompanied by changes in the levels of other synaptic proteins, including the postsynaptic scaffolding protein DLG (*Figure 1—figure supplement 1*). Those results suggested that DNlg1 and DNlg2 promote F-actin assembly at NMJs.

Because the *dnlg1* and *dnlg2* mutants both showed similar defects in postsynaptic F-actin organization, we attempted to determine the relationship between DNlg1 and DNlg2 in the regulation of F-actin assembly by generating *dnlg1; dnlg2* double mutants. We found that these double mutants showed defects similar to those of the *dnlg1* single mutants: the amount of F-actin/α-Spectrin and the relative F-actin area were declined compared with those in the WT but were comparable to those in the *dnlg1* single mutants (*Figure 1D,E and F*). Those results suggested that DNlg1 and DNlg2 regulate postsynaptic F-actin assembly, possibly through a shared pathway.

## DNlg1 and DNlg2 mediate the effect of DNrx on postsynaptic actin

Neurexins are the presynaptic partners of neuroligins. *Drosophila* has only one genuine neurexin homolog: *dnrx* (*Li et al., 2007*; *Zeng et al., 2007*). Previous work has shown that stable expression of DNrx and DNlg1 at the synapse is dependent on the interaction between those two proteins. Thus, in *dnrx* or *dnlg1* mutants, the localization and level of DNlg1 or DNrx, respectively, are significantly impaired (*Banovic et al., 2010*; *Owald et al., 2012*; *Banerjee et al., 2017*). Furthermore, genetic assays indicated that DNrx and DNlg2 regulate bouton growth through a shared pathway (*Chen et al., 2012*). We therefore examined whether DNrx also plays a role in postsynaptic actin regulation. To that end, we analyzed postsynaptic F-actin in *dnrx* mutants and found that both the amount and the distribution area of F-actin and α-Spectrin were significantly decreased in the mutants (*Figure 2A,B,G and H*). Those changes could be rescued by presynaptic expression, but not postsynaptic expression, of DNrx (*Figure 2C,D,G and H*).

The similar changes in postsynaptic F-actin in *dnrx*, *dnlg1*, and *dnlg2* mutants suggested that the postsynaptic actin cytoskeleton is regulated by the trans-synaptic interaction between DNrx and DNlg1 and/or that between DNrx and DNlg2. To test that possibility, we generated *dnlg1; dnrx* and *dnlg2; dnrx* double mutants. The double mutants showed degrees of F-actin and α-Spectrin defects similar to those in the *dnrx*, *dnlg1*, and *dnlg2* single mutants (*Figure 2E–H*). Those results meant

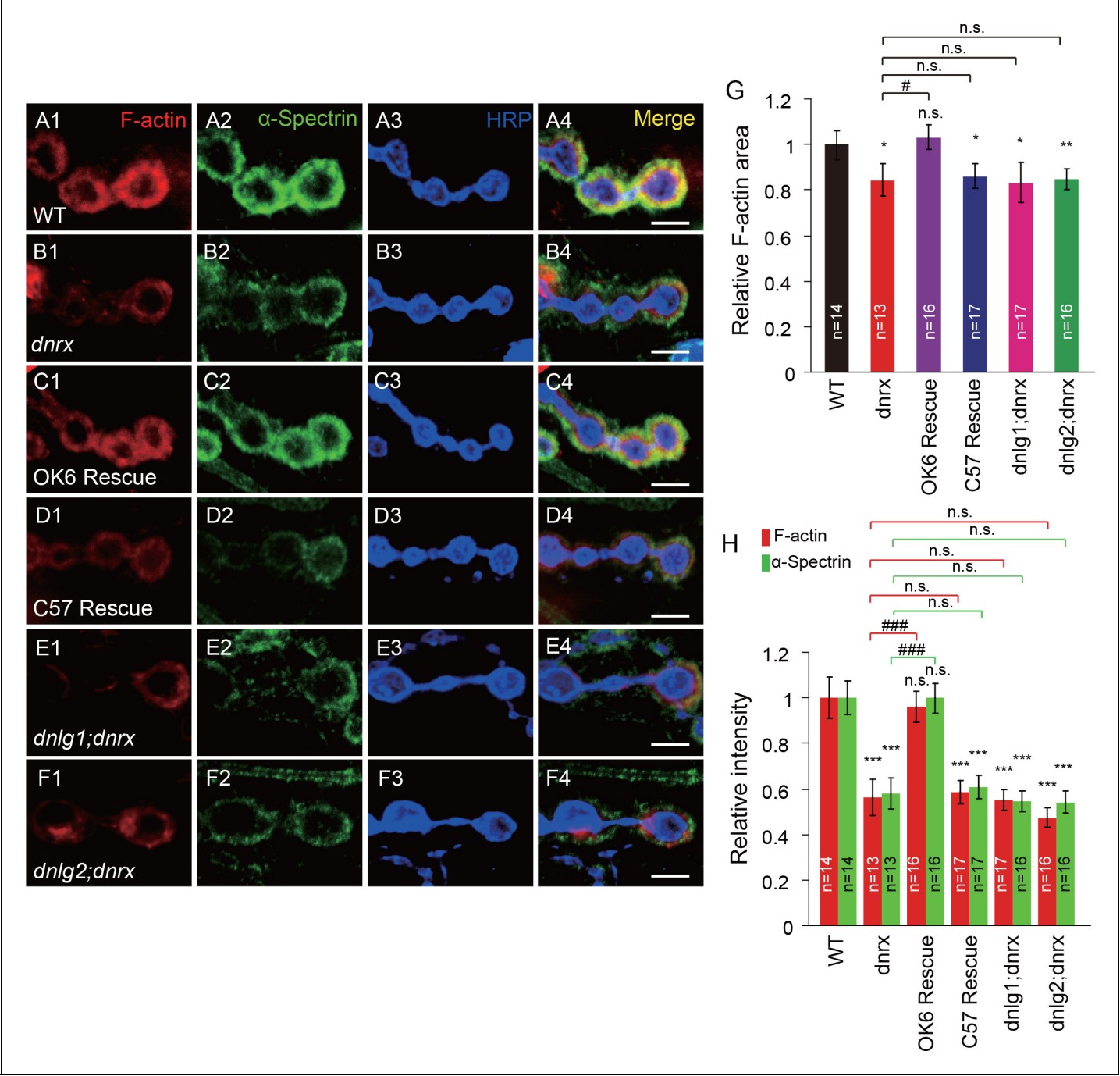

**Figure 2.** DNlg1 and DNlg2 mediate the effect of DNrx on postsynaptic F-actin. (A–F) Confocal images of (A) WT, (B) *dnrx* mutant (*dnrx^{83/174}*), (C) OK6 rescue (*OK6-Gal4/UAS-DNrx;dnrx^{83/174}*), (D) C57 rescue (*UAS-DNrx/+,dnrx^{174}/dnrx^{83},C57-Gal4*), (E) *dnlg1; dnrx* double mutant (*dnrx^{83},dnlg1^{ex2.3}/dnrx^{174}, dnlg1^{ex1.9}*), and (F) *dnlg2; dnrx* double mutant (*dnlg2^{KO70/K070};dnrx^{83/174}*) third instar larvae NMJ type Ib boutons at muscles 12/13 labeled with Texas Red phalloidin (red), anti-α-Spectrin (green), and anti-HRP (blue). (G) Summary graph showing a significant decrease in the relative F-actin area in the *dnrx* mutants, C57 rescue, *dnlg1; dnrx* double mutants, and *dnlg2; dnrx* double mutants compared with the WT. (H) Summary graph showing a significant decrease in the relative fluorescence intensity of F-actin and α-Spectrin in the *dnrx* mutants, C57 rescue, *dnlg1; dnrx* double mutants, and *dnlg2; dnrx* double mutants compared with the WT. The data in (G) and (H) are shown as the mean ± SEM; n represents the number of boutons analyzed; asterisks indicate significant differences between the WT and the indicated genotypes. *p<0.05; **p<0.01; ***p<0.001; n.s., not significant. Hashes indicate significant differences between two indicated genotypes, #p<0.05; ###p<0.001; n.s., not significant. Scale bars: (A–F) 5 μm.
DOI: https://doi.org/10.7554/eLife.30457.006

The following source data is available for figure 2:

*Figure 2 continued on next page*

*Figure 2 continued*

**Source data 1.** Sample size (n), mean, SEM, and one-way ANOVA (and nonparametric) with Tukey's multiple comparisons test are presented for the data in *Figure 2G and H*.

DOI: https://doi.org/10.7554/eLife.30457.007

that presynaptic DNrx regulates the postsynaptic actin cytoskeleton via its interaction with DNlg1 and/or DNlg2.

## DNlg1, but not DNlg2 or DNlg3, binds to the WRC via its conserved WIRS motif

Recent work on the regulation of WAVE function indicates that the intracellular tails of diverse membrane proteins, including protocadherins, G-protein-coupled receptors, netrin receptors, and ion channels, that contain the WIRS consensus motif can bind to the surface of the WRC to regulate the actin cytoskeleton (*Chen et al., 2014*). That binding property is conserved in metazoans, including *Caenorhabditis elegans*, *Drosophila*, and mammals. Using the conserved WIRS sequence, Φ-x-T/S-F-x-x (Φ = preference for bulky hydrophobic residues; x = any residue), we identified a potential WIRS motif (FQSFEA) at the cytoplasmic tail of DNlg1 (*Figure 3A* and *Figure 3—figure supplement 1*). We did not find any such motifs in the cytoplasmic tails of the other *Drosophila* neuroligins (*Figure 3—figure supplement 1*).

To determine whether the cytoplasmic tail of DNlg1 can bind to the WRC, we performed GST pull-down assays by incubating GST proteins fused to the cytoplasmic tail of DNlg1, DNlg2, or DNlg3 with *Drosophila* brain-protein lysates. As shown in *Figure 3B*, the GST fusion protein containing the cytoplasmic tail of DNlg1, but not those containing the tail of DNlg2 or DNlg3, was able to pull down the WRC component of the lysates. To further confirm that the WIRS motif in DNlg1 was responsible for the interaction with the WRC, we generated two mutant fusion proteins with altered WIRS motifs: GST-DNlg1-C$^{F-A}$ and GST-DNlg1-C$^{SF-AA}$. As shown in *Figure 3C*, neither of the mutant fusion proteins could pull down the WRC, indicating that the WIRS motif in the DNlg1 cytoplasmic tail is indispensable for DNlg1 to interact with the WRC.

## DNlg1 recruits WRC to the postsynaptic membrane

DNlg1 is a postsynaptic membrane protein, so the next question was whether WRC could be recruited to the postsynaptic membrane via its interaction with DNlg1. To address that question, we first sought to determine the WRC expression pattern at the NMJ. We used an antibody against SCAR, a key WRC component, to label the NMJ boutons. As shown in *Figure 3D*, bouton-specific staining was clearly visible. To determine whether WRC is expressed in the presynaptic site and/or the postsynaptic site, we utilized the Gal4/upstream activating sequence (UAS) system in combination with a transgene-mediated RNA interference (RNAi) method to knock down endogenous SCAR specifically in either presynaptic cells or postsynaptic cells. As shown in *Figure 3E,F and I*, presynaptic knockdown by OK6-Gal4 did not alter the intensity of SCAR staining, whereas postsynaptic knockdown by C57-Gal4 dramatically diminished the SCAR signal, indicating that the majority, if not all, of SCAR is localized in the postsynaptic site.

After establishing the synaptic localization of SCAR, we analyzed SCAR expression in the absence of DNlg1. As shown in *Figure 3G and I*, SCAR expression was significantly reduced in *dnlg1* mutants. In addition, the distribution of SCAR was disrupted in the mutants so that it appeared dispersed in the postsynaptic cells and no longer accumulated around the boutons (*Figure 3G*, yellow arrow heads), in clear contrast to that in the WT control (*Figure 3D*), indicating that DNlg1 is required for WRC to cluster at the postsynaptic site/membrane. We examined the SCAR distribution in a *dnlg2* mutants and found that the *dnlg2* mutants also had reduced SCAR expression (*Figure 3H and I*). In contrast to that in the *dnlg1* mutants, the majority of remaining SCAR in the *dnlg2* mutants still tightly surrounded the bouton (*Figure 3H*), suggesting that the loss of DNlg2 mainly affected the WRC expression level, but not the WRC distribution.

To further confirm the role of DNlg1 in the membrane recruitment of WRC, we expressed the full-length EGFP-tagged DNlg1 in Schneider 2 (S2) cells and analyzed the subcellular distribution of SCAR. As shown in *Figure 3J*, in nontransfected control cells, SCAR was distributed uniformly

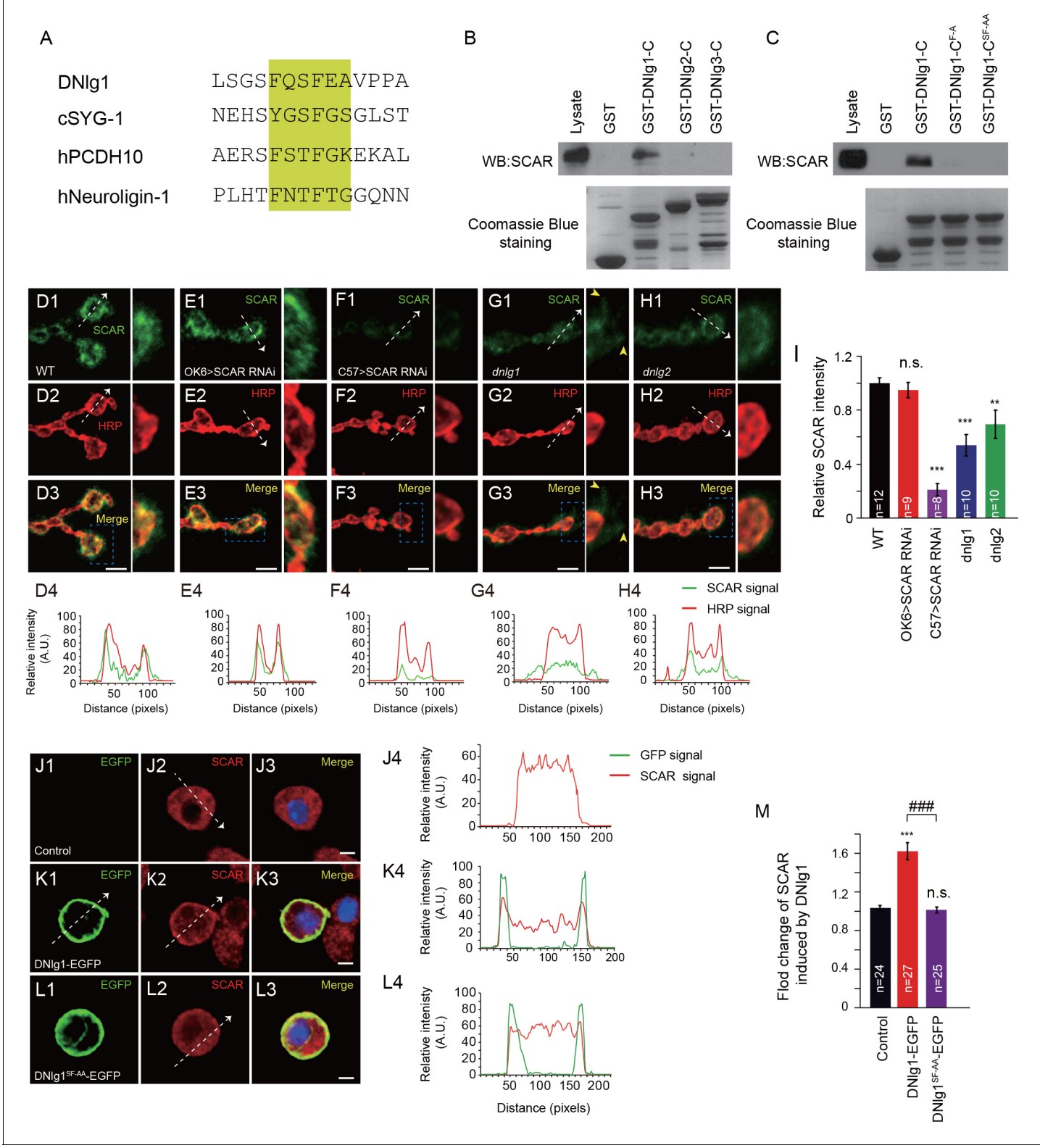

**Figure 3.** DNlg1, but not DNlg2 or DNlg3, binds to the WRC through its WIRS motif and recruits the WRC to the postsynaptic membrane. (**A**) Amino acid sequence alignment of DNlg1, *C. elegans* SYG-1 (cSYG-1), human PCDH10 (hPCDH10), and human Neuroligin 1 (hNeuroligin-1). The WIRS motifs are highlighted in yellow. (**B**) Lysates of WT fly brains were incubated with GST and GST-fused cytoplasmic domains of DNlg1 (GST-DNlg1-C), DNlg2 (GST-DNlg2-C), and DNlg3 (GST-DNlg3-C), respectively. The proteins pulled down with the GST or GST-fused proteins were analyzed by western blots

*Figure 3 continued on next page*

*Figure 3 continued*

using anti-SCAR antibody. Endogenous WRC was retained from the protein lysate by immobilized GST-DNlg1-C, but not by GST, GST-DNlg2-C, or GST-DNlg3-C. GST and GST-fused proteins were detected by Coomassie Blue staining. (C) Western blots showing that endogenous WRC was retained from fly-head lysates by immobilized GST-DNlg1-C, but not by the mutant forms of the WIRS motif GST-DNlg1-C$^{F-A}$ (F changed to A at the first amino acid residue in the WIRS motif) and GST-DNlg1-C$^{SF-AA}$ (SF changed to AA at the third and fourth amino acid residues in the WIRS motif). GST and GST-fused proteins were detected by Coomassie Blue staining. (D–H) Confocal images of (D) WT, (E) OK >SCAR RNAi (*OK6-Gal4/+; UAS-SCAR RNAi/+*), (F) C57 >SCAR RNAi (*C57-Gal4/+; UAS-SCAR RNAi/+*), (G) *dnlg1* mutant (*dnlg1$^{ex1.9/ex2.3}$*), and (H) *dnlg2* mutant (*dnlg2$^{KO70/KO17}$*) third instar larvae NMJ type Ib boutons labeled with anti-SCAR (green) and anti-HRP (red). The majority of SCAR protein was docked on the postsynaptic site. The loss of DNlg1 or DNlg2 caused a reduction in the level of SCAR. Line profile analyses show the distribution and intensity fluctuation of SCAR and HRP in each genotype. Dotted white lines indicate the regions analyzed in the line profile analysis. The directions are indicated by white arrows. Yellow arrow heads indicate diffused SCAR immunostaining signals in the *dnlg1* mutants. (I) Summary graphs of the relative fluorescence intensity of SCAR signals in the indicated genotypes. (J–L) Low-density culture of (J) control S2 cells and (K) cells expressing DNlg1-EGFP or (L) the WIRS mutant form SF-AA (DNlg1$^{SF-AA}$–EGFP) labeled with anti-SCAR and anti-GFP. SCAR was distributed throughout the cytoplasm of control S2 cells but was highly enriched at or near the cell membrane in the cells expressing DNlg1-EGFP, but not in those expressing DNlg1$^{SF-AA}$-EGFP. Line profile analyses show that WT DNlg1, but not DNlg1$^{SF-AA}$, induced the recruitment of SCAR to the plasma membrane and caused the co-localization of DNlg1 and SCAR. Dotted white lines indicate the regions analyzed in the line profile analysis. The directions are indicated by white arrows. (M) The ratio of SCAR intensity at the plasma membrane to that within the cytoplasm was calculated to indicate the recruitment of SCAR to the plasma membrane. A summary graph shows that WT DNlg1, but not DNlg1$^{SF-AA}$, induced the recruitment of SCAR to the plasma membrane. The data in (I) and (M) are shown as the mean ± SEM; n represents the number of boutons analyzed; asterisks indicate significant differences between the WT and the indicated genotypes. **$p<0.01$; ***$p<0.001$; n.s., not significant. Hashes indicate significant differences between two indicated genotypes, ###$p<0.001$. Scale bars: (D–H) 5 μm, (J–L) 5 μm. A. U., artificial unit.

DOI: https://doi.org/10.7554/eLife.30457.008

The following source data and figure supplements are available for figure 3:

**Source data 1.** Sample size (n), mean, SEM, and one-way ANOVA (and nonparametric) with Tukey's multiple comparisons test are presented for the data in *Figure 3I and M*.
DOI: https://doi.org/10.7554/eLife.30457.011

**Figure supplement 1.** The WIRS motif is prevalent in neuroligins.
DOI: https://doi.org/10.7554/eLife.30457.009

**Figure supplement 2.** Examination of SCAR expression and distribution in S2 cells by siRNA.
DOI: https://doi.org/10.7554/eLife.30457.010

**Figure supplement 2—source data 1.** Sample size (n), mean, SEM, and one-way ANOVA (and nonparametric) Turkey's multiple comparisons test are presented for the data in *Figure 3—figure supplement 2B and D*.
DOI: https://doi.org/10.7554/eLife.30457.012

throughout the cytoplasm. That expression pattern was validated by SCAR siRNA experiments (*Figure 3—figure supplement 2*). In EGFP-tagged DNlg1-transfected cells, SCAR tended to be concentrated more at the plasma membrane than in the cytoplasm (*Figure 3K and M*). The expression of mutant DNlg1 [DNlg1$^{SF-AA}$-EGFP (the SF-AA mutation within the WIRS motif)] in S2 cells failed to efficiently recruit WRC to the cell membrane (*Figure 3L and M*) as what WT DNlg1 did. Those results are consistent with the hypothesis that the recruitment of WRC to the postsynaptic membrane by DNlg1 is mediated by a direct interaction between the WRC and DNlg1.

## Interaction between DNlg1 and the WRC is required for postsynaptic F-actin assembly

Although it is widely accepted that the WRC is one of the key players in F-actin assembly, the *in vivo* roles of the WRC in postsynaptic F-actin assembly and NMJ development are still unclear. As a SCAR-null mutant died in the embryonic stages (*Zallen et al., 2002*), we addressed the *in vivo* roles of the WRC using SCAR RNAi lines driven by muscle-specific Gal4. We examined the RNAi efficiency using western blots (*Figure 4A*). The postsynaptic knockdown of SCAR with 24B-Gal4, a strong muscle-specific-Gal4, greatly impaired postsynaptic F-actin assembly and bouton growth (*Figure 4B–H*), suggesting that the WRC plays important roles in postsynaptic F-actin assembly and bouton growth in the *Drosophila* NMJ.

Because DNlg1 is required for membrane recruitment of the WRC, and the WRC is crucial for actin assembly, it is reasonable to hypothesize that the WRC mediates the regulation of postsynaptic F-actin assembly by DNlg1. To test that hypothesis, we performed the following rescue experiments. In *dnlg1* mutants, the distributions of both the WRC and postsynaptic F-actin were defective

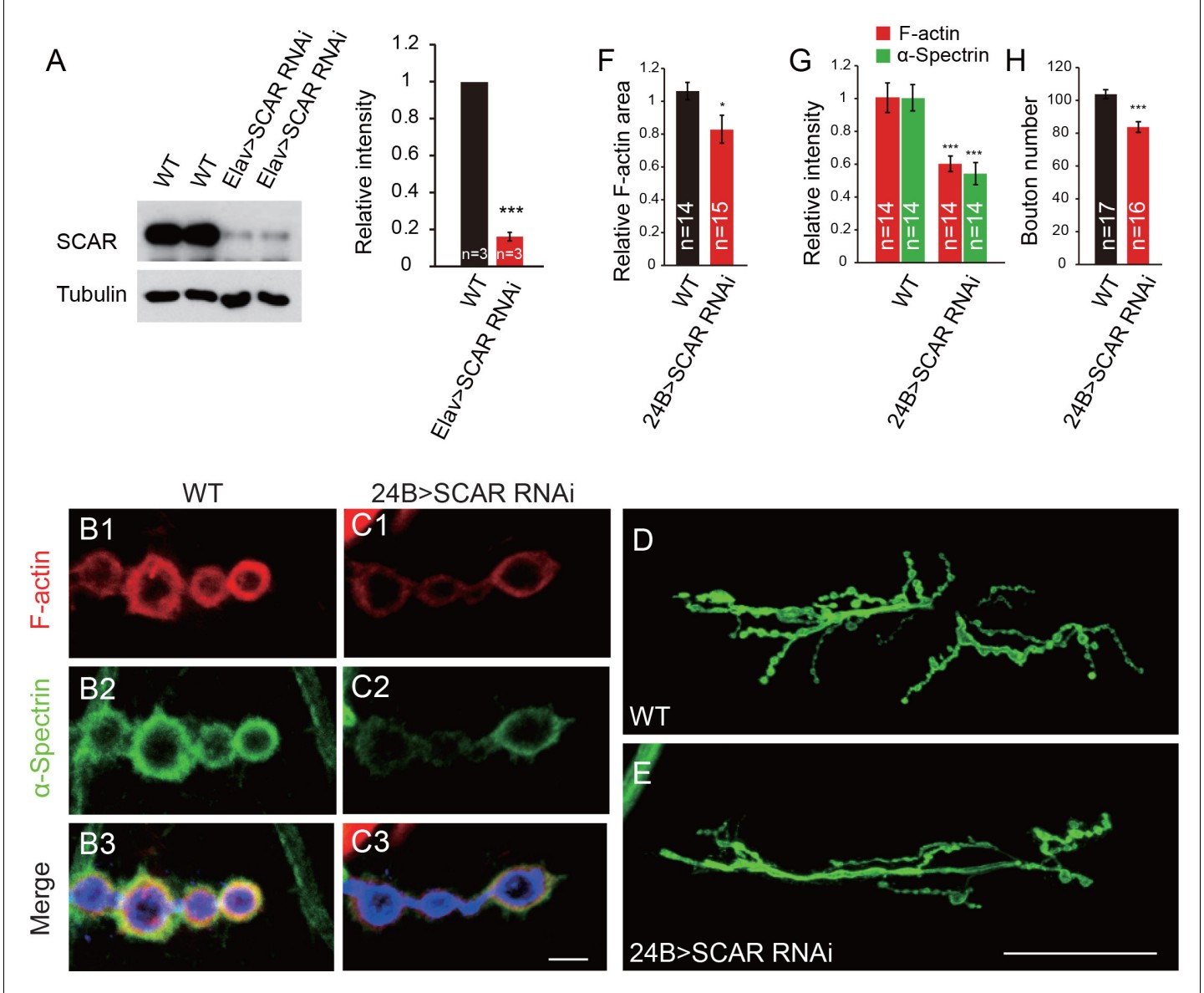

**Figure 4.** Postsynaptic knockdown of SCAR impairs F-actin assembly and inhibits bouton growth. (**A**) Brain lysates from WT and Elav > SCAR RNAi (*Elav/+; UAS-SCAR RNAi/+*) flies were subjected to western blots with anti-SCAR antibody. SCAR expression was dramatically inhibited in the Elav > SCAR RNAi line. A summary graph shows the relative SCAR intensity in both lines. (**B–C**) Confocal images of (**B**) WT and (**C**) 24B > SCAR RNAi (*UAS-SCAR RNAi/+; 24B-Gal4/+*) third instar larvae NMJ type Ib boutons at muscles 12/13 labeled with Texas Red phalloidin (red), anti-α-Spectrin (green), and anti-HRP (blue). (**D–E**) Confocal images of WT and 24B > SCAR RNAi third instar larvae NMJs at muscles 6/7 segment A2 labeled with anti-HRP. (**F–H**) Quantitative analysis of the relative F-actin area, intensity of F-actin and α-Spectrin, and bouton number in WT and 24B > SCAR RNAi flies. The data in (**A**) and (**F–H**) are shown as the mean ± SEM; n represents the number of replicate experiments in (**A**), the number of boutons analyzed in (**F**) and (**G**), and the number of NMJs analyzed in (**H**); asterisks indicate significant differences between the WT and the indicated genotypes. *p<0.05; ***p<0.001. Scale bars: (**B, C**) 5 μm, (**D, E**) 50 μm.

DOI: https://doi.org/10.7554/eLife.30457.013

The following source data is available for figure 4:

**Source data 1.** Sample size (n), mean, SEM, and Mann–Whitney test or Student's t-test for *Figures 4A, F, G and H*.

DOI: https://doi.org/10.7554/eLife.30457.014

compared with those in WT animals (*Figure 5A,B,E and F*). The postsynaptic expression of a transgene containing the full-length DNlg1 (rescue-WT) completely rescued those defects in the *dnlg1* mutants (*Figures 5C,G and I–K*). However, the postsynaptic expression of mutant DNlg1 [DNlg1^SF-

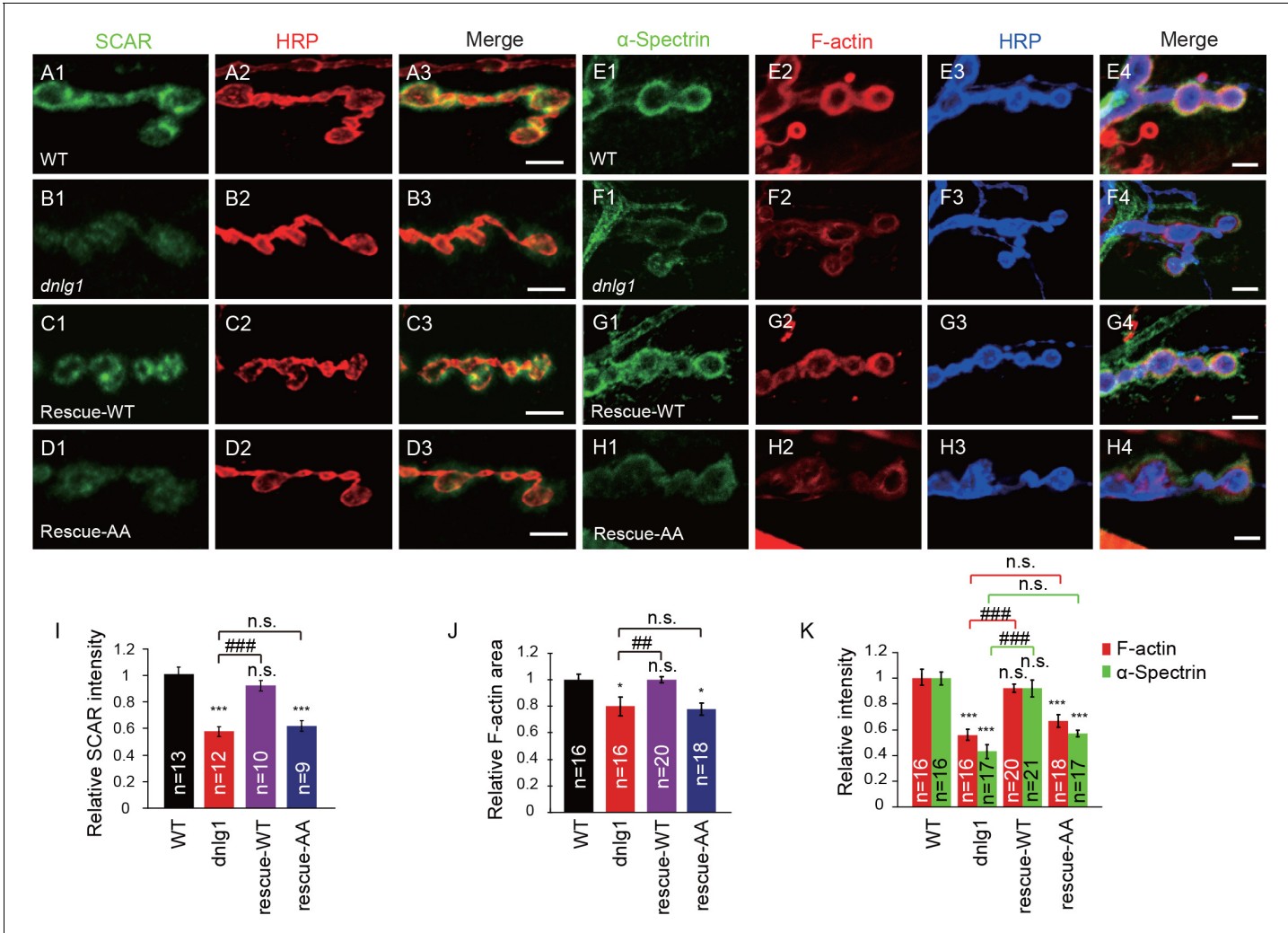

**Figure 5.** The interaction between DNlg1 and the WRC is required for WRC localization and F-actin assembly. (A–D) Confocal images of third instar larvae NMJ type Ib boutons double labeled with anti-SCAR (green) and anti-HRP (red). (A) WT. (B) *dnlg1* mutant (*dnlg1*<sup>ex1.9/ex2.3</sup>). (C) Muscle-specific rescue line using Mef2-Gal4 driving a full-length DNlg1 coding sequence in a *dnlg1* mutant (*Mef2-Gal4/+; dnlg1*<sup>ex1.9</sup>, *UAS-DNlg1/dnlg1*<sup>ex2.3</sup>; rescue-WT). (D) Muscle-specific line using Mef2-Gal4 driving a full-length DNlg1 coding sequence containing a WIRS mutation in a *dnlg1* mutant (*Mef2-Gal4/+; dnlg1*<sup>ex1.9</sup>, *UAS-DNlg1*<sup>SF-AA</sup>/*dnlg1*<sup>ex2.3</sup>; rescue-AA). The altered SCAR distribution and protein level in the *dnlg1* mutants were rescued in the rescue-WT line, but not in the rescue-AA line. (E–H) Confocal images of (E) WT, (F) *dnlg1* mutant, (G) rescue-WT, and (H) rescue-AA third instar larvae NMJ type Ib boutons at muscles 12/13 labeled with Texas Red phalloidin (red), anti-α-Spectrin (green) and anti-HRP (blue). The impaired F-actin and α-Spectrin cytoskeleton assemblies in the *dnlg1* mutants were rescued in the rescue-WT line, but not in the rescue-AA line. (I) A summary graph showing that the reduced relative SCAR fluorescence intensity in the *dnlg1* mutants was rescued in the rescue-WT line, but not in the rescue-AA line. (J, K) Summary graphs of the relative F-actin area and the relative fluorescence intensity of F-actin and α-Spectrin in the indicated genotypes. The data shown in (I–K) are mean ± SEM; n represents the number of boutons analyzed; asterisks indicate significant differences between the WT and the indicated genotypes. *p<0.05; ***p<0.001. n.s., not significant. Hashes indicate significant differences between two indicated genotypes, ##p<0.01; ###p<0.001; n.s., not significant. Scale bars: (A–D) 5 μm; (E–H) 5 μm.

DOI: https://doi.org/10.7554/eLife.30457.015

The following source data and figure supplement are available for figure 5:

**Source data 1.** Sample size (n), mean, SEM, and one-way ANOVA (and nonparametric) with Tukey's multiple comparisons test are presented for the data in *Figure 5I*.

DOI: https://doi.org/10.7554/eLife.30457.017

**Figure supplement 1.** DNlg1<sup>SF-AA</sup>, the mutant form of DNlg1, can be normally targeted to postsynaptic sites.

DOI: https://doi.org/10.7554/eLife.30457.016

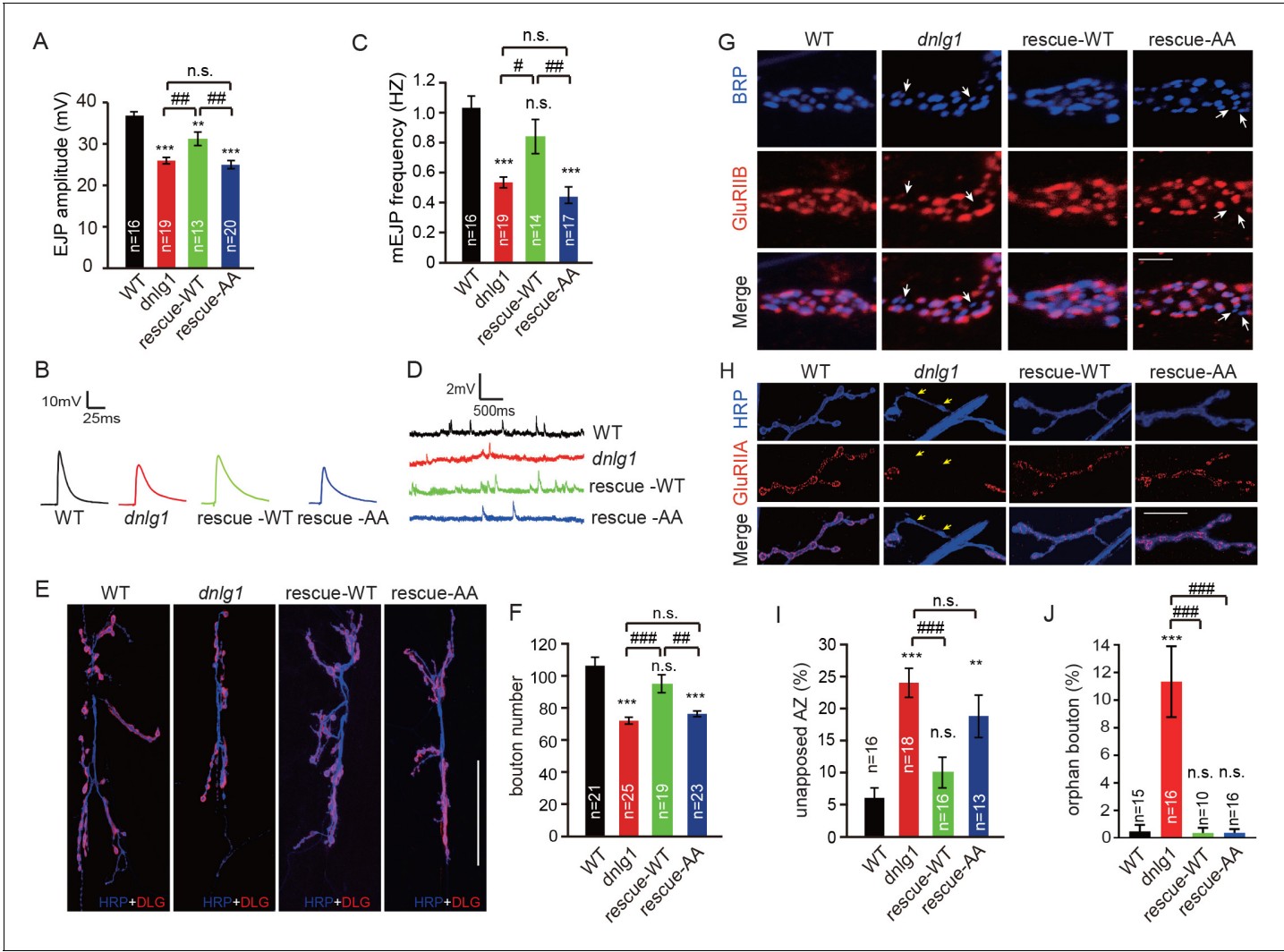

**Figure 6.** The interaction between DNlg1 and the WRC is critical for the functions of DNlg1 in bouton growth, synaptic transmission, and GluR recruitment. (A–D) Electrophysiological analysis of control and *dnlg1* mutant NMJs on muscles 6/7 of third instar larvae indicated that postsynaptic expression of WT DNlg1, but not mutant DNlg1^SF-AA, rescued the reduced synaptic transmission in the *dnlg1* mutants. (A) Bar graphs of the mean EJP (in mV) and (B) representative traces of the EJP amplitudes in WT, *dnlg1* mutant (*dnlg1^ex1.9/ex2.3*), rescue-WT (*Mef2-Gal4/+; dnlg1^ex1.9/dnlg1^ex2.3, UAS-DNlg1-EGFP*), and rescue-AA (*Mef2-Gal4/+; dnlg1^ex1.9/dnlg1^ex2.3, UAS-DNlg1^SF-AA-EGFP*) NMJs. (C) Bar graphs of the mean mEJP frequency and (D) representative traces of the mEJP frequency in WT, *dnlg1* mutant, rescue-WT, and rescue-AA NMJs. (E–F) Postsynaptic expression of WT DNlg1, but not mutant DNlg1^SF-AA, could rescue the reduced bouton number in the *dnlg1* mutants. (E) Confocal images of WT, *dnlg1* mutant, rescue-WT, and rescue-AA third instar larvae NMJ type Ib boutons at muscles 6/7 labeled with DLG (red) and anti-HRP (blue). (F) Quantitative data of bouton numbers in (E). (G) Confocal images of third instar larvae NMJ type Ib boutons at muscle four labeled with GluRIIB (red) and postsynaptic active zone marker Brp (blue) to quantify GluR unaligned with the active zone in the same four lines. (H) Confocal images of third instar larvae NMJ type I boutons at muscle four labeled with GluRIIA (red) and HRP (blue) to quantify orphan boutons in the same four lines. (I and J) Statistical analysis of (I) unopposed active zone and (J) orphan boutons that have a presynaptic (HRP) site but lack postsynaptic GluR localization. The green fluorescence signal indicates the expression of EGFP. White arrow: unopposed GluRIIB and BRP; yellow arrow: orphan boutons completely losing GluRIIA positioning. The data in (A), (C), (F), (I), and (J) are shown as the mean ± SEM; n represents the number of NMJ samples analyzed; asterisks indicate significant differences between the WT and the indicated genotypes. **p<0.01; ***p<0.001; n.s., not significant. Hashes indicate significant differences between genotypes. #p<0.05; ##p<0.01; ###p<0.001; n.s., not significant. Scale bars: (E) 50 μm, (G) 1 μm, (H) 20 μm.

DOI: https://doi.org/10.7554/eLife.30457.018

The following source data and figure supplements are available for figure 6:

**Source data 1.** Sample size (n), mean, SEM, and one-way ANOVA (and nonparametric) with Tukey's multiple comparisons test are presented for the data in *Figures 6A, C, F, I and J*.

DOI: https://doi.org/10.7554/eLife.30457.020

**Figure supplement 1.** Protein levels of GluRIIA and GluRIIB appear normal in *dnlg1* mutants and rescue lines.

*Figure 6 continued on next page*

*Figure 6 continued*

DOI: https://doi.org/10.7554/eLife.30457.019

**Figure supplement 1—source data 1.** Sample size (n), mean, SEM, one-way ANOVA (and nonparametric) with Tukey's multiple comparisons test are presented for the data in *Figure 6—figure supplement 1B and D*.

DOI: https://doi.org/10.7554/eLife.30457.021

AA (rescue-AA)] failed to rescue the defects (*Figures 5D,H and I–K*). The mutant DNlg1 could target the postsynaptic membrane as well as the WT DNlg1 (*Figure 5—figure supplement 1*), indicating that the WIRS motif is not required for the synaptic targeting of DNlg1 itself. Therefore, the interaction between DNlg1 and the WRC is specifically required for the postsynaptic localization of the WRC and subsequent F-actin regulation.

## Interaction between DNlg1 and the WRC is essential for the roles of DNlg1 in normal NMJ growth, synaptic transmission, and GluR recruitment

DNlg1 is known to be a key factor in the regulation of bouton growth and synaptic transmission (*Banovic et al., 2010*). We asked whether those processes require the interaction between DNlg1 and WRC. To address that question, we performed electrophysiological recording and morphological analyses of *dnlg1* mutants and also rescue experiments in which WT DNlg1 or mutant DNlg1$^{SF-AA}$ was postsynaptically expressed in *dnlg1* mutants. Consistent with previous studies (*Banovic et al., 2010*), the *dnlg1* mutants displayed declined excitatory junction potential (EJP) amplitude (*Figure 6A and B*), lower miniature excitatory junction potential (mEJP) frequency (*Figure 6C and D*), and less NMJ bouton numbers (*Figure 6E and F*) compared with WT flies. The postsynaptic expression of WT DNlg1 in the *dnlg1* mutants (rescue-WT) could rescue those phenotypes to levels comparable to those in WT flies (*Figure 6A–F*). The postsynaptic expression of the DNlg1$^{SF-AA}$ (rescue-AA) failed to reverse the deficits in the *dnlg1* mutants (*Figure 6A–F*), suggesting that the interaction between DNlg1 and the WRC is required for the functions of DNlg1 in synaptic transmission and bouton growth.

The postsynaptic recruitment of GluRs is abnormal in *dnlg1* mutants, resulting in misapposition between postsynaptic GluRs and presynaptic active zones (*Banovic et al., 2010*). Consistent with that phenomenon, we observed that compared with that in WT flies, around 15–20% more of the presynaptic active zone [bruchpilot (BRP)-stained area] in *dnlg1* mutants lacked the proper alignment with postsynaptic GluRs (*Figure 6G*). A small portion of boutons in the *dnlg1* mutants (around 10%) had a more severe phenotype, called orphan bouton, in which the postsynaptic GluRs were lost entirely (*Figure 6H*). As expected, the postsynaptic expression of WT DNlg1 with muscle-specific Mef2-Gal4 rescued those defects in the *dnlg1* mutants (*Figure 6G–J*); however, the postsynaptic expression of mutant DNlg1 (DNlg1$^{SF-AA}$) could only partially rescue the defects (*Figure 6G–J*). That means that the abnormal GluR recruitment in the *dnlg1* mutants was at least partially due to the abolished interaction between DNlg1 and the WRC. Although there were differences in GluR recruitment among the studied lines, the overall expression levels of GluRIIA and GluRIIB were similar among all of the lines (*Figure 6—figure supplement 1*).

Altogether, our results indicate that DNlg1-WRC-mediated postsynaptic F-actin assembly is required for the functions of DNlg1 in bouton growth and synaptic transmission and is at least partially, if not exclusively, responsible for the regulation of postsynaptic GluR recruitment.

## Ectopic expression of DNlg1 at type II boutons is sufficient to promote F-actin assembly

Our experiments so far showed that the DNlg1-WRC interaction is necessary for postsynaptic actin cytoskeleton assembly and normal morphology and synaptic functions at NMJs. We next asked if that interaction is sufficient to promote F-actin assembly. To address that question, we ectopically expressed DNlg1 at type II boutons. Compared with type I boutons (*Figure 7A3*, yellow wedges), type II boutons are typically smaller in size, extend for a longer distance on the muscle fibers, and are connected by a long, thin axonal process (*Figure 7A3*, pink wedges). Indeed, type I and type II boutons form different types of synapses with different presynaptic and postsynaptic properties,

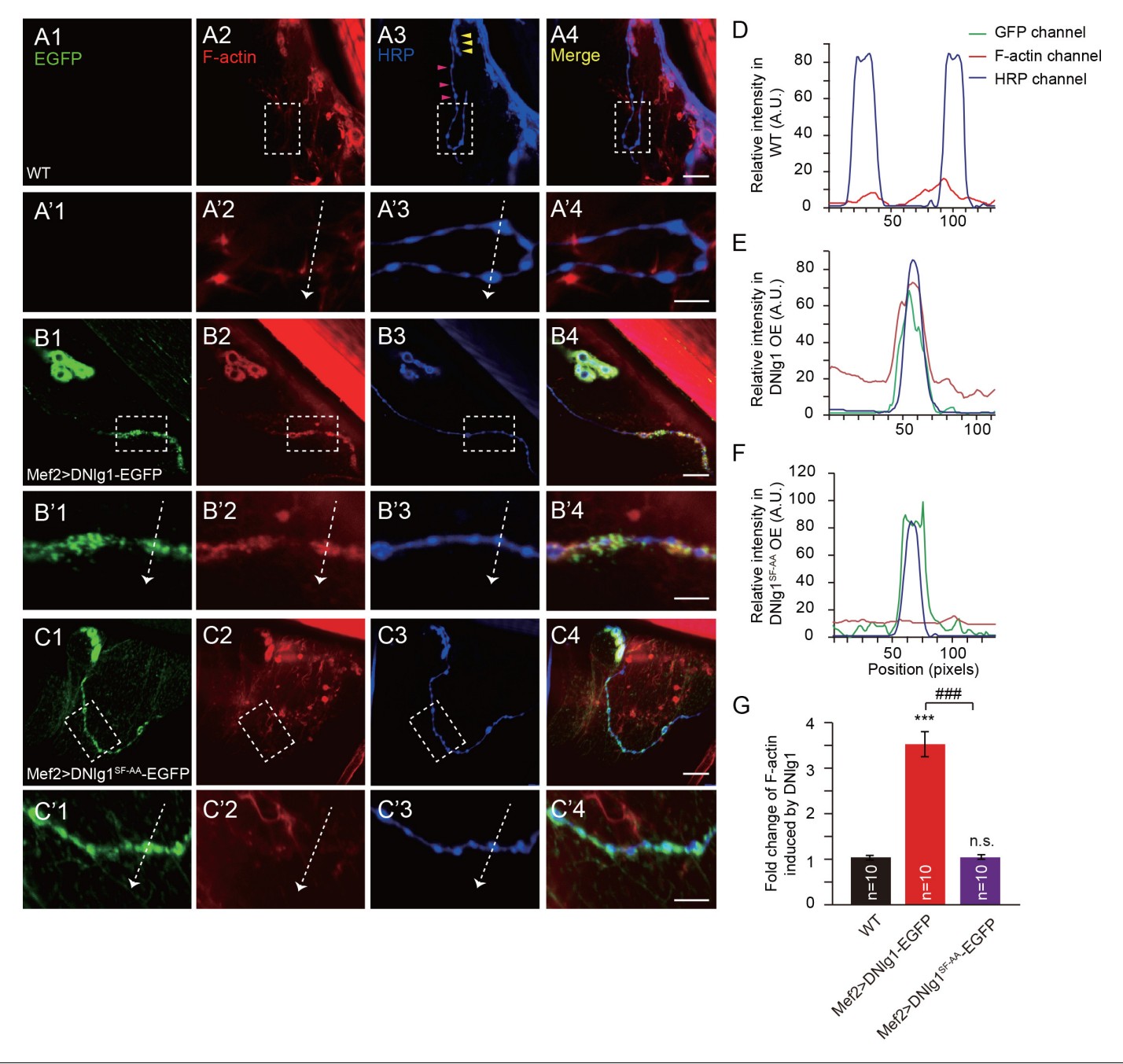

**Figure 7.** Ectopic expression of DNlg1 at type II boutons is sufficient to induce new F-actin assembly. (A–C) Confocal images of (A) WT, (B) Mef2 >DNlg1 EGFP (*Mef2-Gal4/+; UAS-DNlg1-EGFP/+*), and (C) Mef2 >DNlg1$^{SF-AA}$-EGFP (*Mef2-Gal4/+; UAS-DNlg1$^{SF-AA}$-EGFP/+*) third instar larvae NMJ type II boutons at muscles 12/13 labeled with phalloidin (red) and anti-HRP (blue). High-magnification images in A', B', and C' correspond to the view in the dotted rectangles in A, B, and C. The green fluorescence signal indicates the expression of the EGFP protein. The expression of DNlg1-EGFP at the postsynaptic site of type II boutons induced the formation of new F-actin at type II boutons, whereas the expression of DNlg1$^{SF-AA}$-EGFP failed to do so. Yellow wedges in Figure A3 indicate type I bouton, and pink wedges indicate type II boutons. Dotted lines indicate the regions analyzed in (D–F), and the directions are indicated by white arrows. (D–F) Line profile analyses showing that WT DNlg1, but not DNlg1$^{SF-AA}$, induced the formation of F-actin and led to the co-localization of DNlg1 and F-actin at type II boutons. (G) The ratio of F-actin intensity at type II boutons to that of adjacent background was calculated to indicate the recruitment of F-actin at type II boutons by DNlg1. A summary graph shows that WT DNlg1, but not DNlg1$^{SF-AA}$, induced the recruitment of F-actin to type II boutons. The data in (G) are shown as the mean ± SEM; n represents the number of boutons analyzed; asterisks indicate significant differences between the WT and the indicated genotypes. ***p<0.001; n.s., not significant. Hashes indicate significant differences between two indicated genotypes. ###p<0.001. Scale bars: (A–C) 20 µm, (A'–C') 10 µm. A.U., artificial unit.

*Figure 7 continued on next page*

*Figure 7 continued*

DOI: https://doi.org/10.7554/eLife.30457.022

The following source data is available for figure 7:

**Source data 1.** Sample size (n), mean, SEM, and one-way ANOVA (and nonparametric) with Tukey's multiple comparisons test are presented for the data in *Figure 7G*.

DOI: https://doi.org/10.7554/eLife.30457.023

such as the lack of defined SSR structures in type II boutons (*Budnik and Gorczyca, 1992*; *Jia et al., 1993*; *Johansen et al., 1989*). DNlg1 is mainly localized at type I boutons; however, postsynaptic overexpression of DNlg1 has shown that DNlg1 can be ectopically targeted to type II boutons (*Banovic et al., 2010*). In WT flies, we found no significant F-actin labeling at type II boutons (*Figure 7A,D*). When we expressed DNlg1-EGFP with the muscle-specific Gal4 driver (Mef2-Gal4) in a WT background, we found marked F-actin staining at type II boutons that was similar to that observed at type I boutons (*Figure 7B,E and G*). Consistent with those biochemical data, mutant DNlg1 with an altered WIRS motif failed to induce F-actin assembly (*Figure 7C,F and G*). In addition, we found WRC clustered at type II boutons in DNlg1-EGFP transgenic lines (*Figure 8B,E and G*) but not in the WT line or a line expressing mutant DNlg1 with an altered WIRS motif (*Figure 8A,C,D,F and G*). Those results suggest that DNlg1 is sufficient to induce F-actin formation at the postsynaptic membrane and that the process requires interaction between DNlg1 and the WRC.

## Discussion

It is generally agreed that the trans-synaptic complex formed by neurexin–neuroligin is critically involved in synapse development and function. A major unresolved issue is how that trans-synaptic interaction regulates those processes. In this study, we revealed a novel mechanism mediated by the actin cytoskeleton. DNlg1 directly interacts with and recruits the WRC, a potent regulator of actin reorganization, to the postsynaptic membrane. That interaction is both necessary and sufficient for DNlg1 to assemble actin filaments. Furthermore, the DNlg1-WRC interaction is essential for DNlg1 to regulate both synaptic morphology and synaptic function. Those findings establish a key role for actin in mediating the effects of neurexin–neuroligin at NMJs.

A recent study (*Banerjee et al., 2017*) showed that the DNrx-DNlg1 complex regulates synaptic cytoarchitecture and growth through bone morphogenetic protein (BMP) signaling. According to that conclusion, while DNlg1 may affect actin networks, the effect is not direct, because the BMP signaling regulates actin reorganization via small G proteins (*Ball et al., 2010*). Our study provides compelling evidence that DNlg1 actually is a direct regulator of the actin cytoskeleton through its interaction with the WRC. First, the amount and distribution of postsynaptic F-actin are severely impaired in *dnlg1* mutants. Second, DNlg1 binds to the WRC through the WIRS motif in its cytoplasmic tail, and that binding is crucial for WRC postsynaptic localization. Third, the interaction between DNlg1 and the WRC is required for postsynaptic F-actin assembly. Fourth, ectopic expression of DNlg1 is sufficient to induce postsynaptic F-actin formation, and that effect is dependent on the interaction between DNlg1 and WRC. Finally, the regulation of F-actin assembly by DNlg1 is indispensable for the functions of DNlg1 in bouton growth and synaptic transmission. Postsynaptic expression of WT DNlg1, but not mutant DNlg1[SF-AA], could rescue the defects in *dnlg1* mutants, highly suggesting the necessity of the interaction between DNlg1 and the WRC for those biological events. An interesting observation was that the postsynaptic expression of DNlg1[SF-AA] in *dnlg1* mutants could partially rescue the defects of *dnlg1* mutants in GluR recruitment. That means, if WRC is not the only factor, but at least it is one of the factors to affect DNlg1 mediated GluR recruitment. Other regulators able to interact with DNlg1, which participate into GluR recruitment somehow, wait to be revealed in the near future.

Consistent with the previous finding that DNrx is required for the stabilization of postsynaptic DNlg1 (*Banovic et al., 2010*), *dnrx* mutants showed defects in postsynaptic F-actin similar to those seen in the *dnlg1* mutants. Our results also showed that DNlg2 has a positive effect on F-actin similar to that of DNlg1, but it remains unknown how DNlg2 regulates postsynaptic F-actin assembly, because DNlg2 does not interact with the WRC. Because *dnrx*, *dnlg1*, and *dnlg2* single mutants,

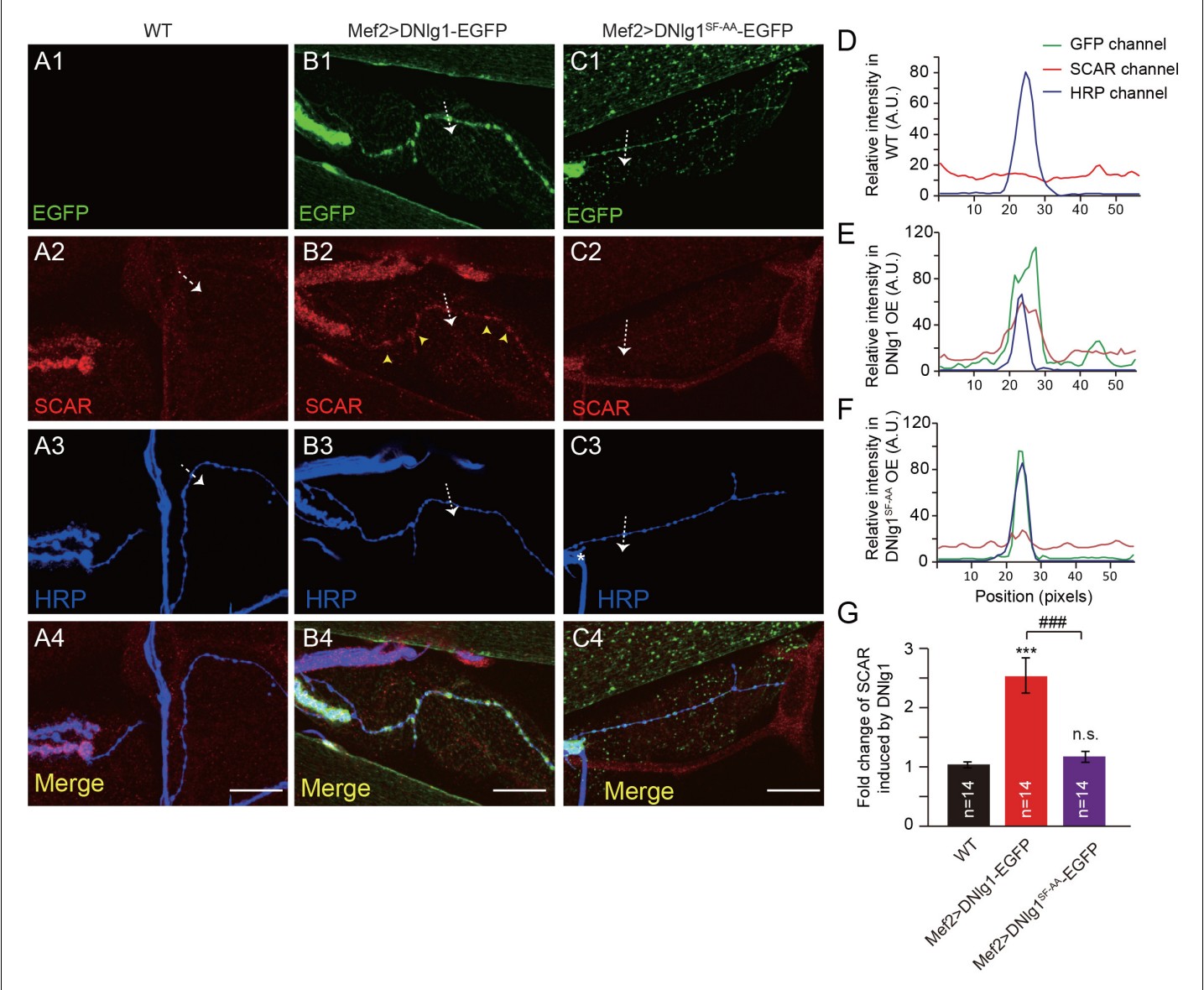

**Figure 8.** Ectopic expression of DNlg1 is sufficient to recruit WRC to type II boutons. (**A–C**) Confocal images of (**A**) WT, (**B**) Mef2 >DNlg1 EGFP (*Mef2-Gal4/+; UAS-DNlg1-EGFP/+*), and (**C**) Mef2>DNlg1$^{SF-AA}$ -EGFP (*Mef2-Gal4/+; UAS-DNlg1$^{SF-AA}$-EGFP/+*) third instar larvae NMJ at muscles 12/13 labeled with anti-SCAR (red) and anti-HRP (blue). The green fluorescence signal indicates the expression of the EGFP protein. The expression of DNlg1-EGFP at the postsynaptic site of type II boutons induced clusters of SCAR (yellow arrow heads in B2), whereas that of DNlg1$^{SF-AA}$ -EGFP did not. Dotted lines indicate the regions analyzed in (**D–F**), and the direction is marked by an white arrows. (**D–F**) Line profile analyses show that WT DNlg1, but not DNlg1$^{SF-AA}$, induced SCAR clusters and co-localization of DNlg1 and SCAR at type II boutons. (**G**) Quantification of SCAR intensity at type II boutons relative to that in the adjacent background. The data in (**G**) are shown as mean ± SEM; n represents the number of boutons; asterisks indicate significant differences between the WT and the indicated genotypes. ***p<0.001; n.s., not significant. Hashes indicate significant differences between two indicated genotypes. ###p<0.001. Scale bars: (**A–C**) 20 μm. A.U., artificial unit. Figure supplements (5).

DOI: https://doi.org/10.7554/eLife.30457.024

The following source data is available for figure 8:

**Source data 1.** Sample size (n), mean, SEM, and one-way ANOVA (and nonparametric) with Tukey's multiple comparisons test are presented for the data in *Figure 8G*.

DOI: https://doi.org/10.7554/eLife.30457.025

*dnrx; dnlg1/2* double mutants, and *dnlg1; dnlg2* double mutants all exhibit similar defects in F-actin, we hypothesize that DNlg1, DNlg2, DNrx, and the WRC may form a large protein complex, so that

single deletions of any of the components of the complex may have similar effects on actin regulation. There are several other possibilities to consider if that hypothesis is not correct. One possibility is that DNlg2 regulates F-actin via p21 protein (Cdc42/Rac)-activated kinase (DPAK), an important regulator of F-actin assembly that has been shown to be reduced in *dnlg2* mutants (*Sun et al., 2011*). Another possibility is that as the PDZ-binding motif of mouse Nlg1 can bind to spine-associated Rap guanosine triphosphatase–activating protein (SPAR) and subsequently activate the LIMK1/cofilin pathway to regulate actin filaments, spine morphology, and plasticity (*Liu et al., 2016*), it is reasonable to ask whether the PDZ-binding motif of DNlg2 can bind to Rap GTPase activating protein 1 (RapGAP1), the *Drosophila* homolog of SPAR. In those contexts, it is important to note that different DNlg mutants exhibit somewhat different defects. The loss of *dnlg1* leads to reduced SSR areas and the disruption of GluR alignment with the presynaptic active zone (*Banovic et al., 2010*). In *dnlg2* mutants, the clustering of GluRIIA is increased, whereas that of GluRIIB and the size of the SSR area are reduced (*Sun et al., 2011*). In *dnlg3* mutants, only the clustering of GluRIIA is reduced (*Xing et al., 2014*). Therefore, it is likely that members of the neuroligin family utilize diverse mechanisms to regulate synapse development and function. It will be important to identify which of those mechanisms involves actin reorganization and how the different mechanisms interact.

Sequence analysis shows that the WIRS motif is prevalent in neuroligins, especially vertebrate neuroligins. Four have been identified in rodents and humans, including a less conserved Nlg4 in rodents (*Südhof, 2008*). Among the neuroligins, Nlg1, Nlg3, and Nlg4 (4X, 4Y) have more amino acid sequence similarity with each other than they do with Nlg2. Biochemistry and immunocytochemistry studies show that all human neuroligins are postsynaptic proteins and that Nlg2 is specifically expressed at inhibitory synapses, whereas Nlg1 and Nlg3 are expressed at excitatory synapses or at both excitatory and inhibitory synapses. All vertebrate neuroligins, except Neuroligin 2, contain the WIRS motif, implying that the regulation of F-actin assembly by neuroligins may determine the specification of the postsynaptic apparatus.

In summary, we identified a novel process by which neuroligins regulate the postsynaptic actin cytoskeleton, which we propose as a key mechanism to mediate the regulatory effect of neuroligins and neurexins on synapse development and function at the *Drosophila* NMJ. We believe that further investigations of the regulation of the postsynaptic cytoskeleton by neuroligins will provide an important avenue of research to dissect the molecular mechanisms governing brain development and function.

# Materials and methods

**Key resources table**

| Reagent type (species) or resource | Designation | Source or reference | Identifiers | Additional information |
|---|---|---|---|---|
| genetic reagent (*D. melanogaster*) | dnlg1[ex1.9] | PMID: 20547130 | | |
| genetic reagent (*D. melanogaster*) | dnlg1[ex2.3] | PMID: 20547130 | | |
| genetic reagent (*D. melanogaster*) | dnlg2[KO70] | PMID: 21228178 | | |
| genetic reagent (*D. melanogaster*) | dnlg2[KO17] | PMID: 21228178 | | |
| genetic reagent (*D. melanogaster*) | dnrx[83] | PMID: 23352167; PMID: 17498701 | | |
| genetic reagent (*D. melanogaster*) | dnrx[174] | PMID: 23352167; PMID: 17498701 | | |
| genetic reagent (*D. melanogaster*) | UAS-DNrx | PMID: 23352167; PMID: 17498701 | | |
| genetic reagent (*D. melanogaster*) | UAS-SCAR RNAi | Center of Biomedical Analysis, Tsinghua University | TH: 02179 .N; RRID: BDSC_51803 | |
| genetic reagent (*D. melanogaster*) | UAS-DNlg1 | lab generated; this paper | | |

*Continued on next page*

*Continued*

| Reagent type (species) or resource | Designation | Source or reference | Identifiers | Additional information |
|---|---|---|---|---|
| genetic reagent (*D. melanogaster*) | UAS-DNlg1$^{SF-AA}$ | lab generated; this paper | | |
| genetic reagent (*D. melanogaster*) | UAS-DNlg1-EGFP | lab generated; this paper | | |
| genetic reagent (*D. melanogaster*) | UAS-DNlg1$^{SF-AA}$-EGFP | lab generated; this paper | | |
| genetic reagent (*D. melanogaster*) | OK6-Gal4 | Bloomington Drosophila Stock center | BDSC: 64199 | |
| genetic reagent (*D. melanogaster*) | Elav-Gal4 | Bloomington Drosophila Stock center | BDSC: 458 | |
| genetic reagent (*D. melanogaster*) | C57-Gal4 | PMID: 8893021 | | |
| genetic reagent (*D. melanogaster*) | 24B-Gal4 | Kyoto Stock center | DGGR:106496 | |
| genetic reagent (*D. melanogaster*) | Mef2-Gal4 | Bloomington Drosophila Stock center | BDSC: 27390 | |
| cell line (*D. melanogaster*) | S2 | China Center for Type Culture Collection, CCTCC | GDC138; RRID:CVCL_Z992 | Improved STR profiling Mycoplasma contamination test: negative |
| antibody | Mouse anti-α-Spectrin | Developmental Studies Hybridoma Bank | DSHB: 3A9; RRID: AB_528473 | 1:50 for IHC |
| antibody | anti-DLG | Developmental Studies Hybridoma Bank | DSHB:4F3; RRID: AB_528203 | 1:50 for IHC |
| antibody | anti-Brp | Developmental Studies Hybridoma Bank | DSHB:nc82; RRID:AB_2314866 | 1:50 for IHC |
| antibody | anti-SCAR | Developmental Studies Hybridoma Bank | DSHB: P1C1-SCAR; RRID:AB_2618386 | 1:50 for IHC, 1:200 for western blotting |
| antibody | rabbit anti-HRP | Jackson ImmunoResearch | Jackson ImmunoResearch: 323-005-021 RRID: AB_2314648 | 1:1000 for IHC |
| antibody | Goat anti-HRP | Jackson ImmunoResearch | Jackson ImmunoResearch: 123-005-021 RRID: AB_2338952 | 1:1000 for IHC |
| antibody | Mouse anti-Tubulin | Sigma | Sigma: Clone DM1A; RRID:AB_477593 | 1:10000 for western blotting |
| antibody | Alexa 488-, 555-, or 637 - secondaries | Molecular Probes | | |
| Other | Texas Red-conjugated Phalloidin | Molecular Probes | ThermoFisher: T7471 | 1:6 for IHC |

## Fly stocks

All flies were reared at 25°C in standard medium unless otherwise specified. The w1118 flies were used as WT controls in this study. The following fly mutants were used: *dnlg1$^{ex1.9}$* and *dnlg1$^{ex2.3}$* (*Banovic et al., 2010*), *dnlg2$^{KO70}$* and *dnlg2$^{KO17}$* (*Sun et al., 2011*), and *dnrx$^{83}$* and *dnrx$^{174}$* (*Tian et al., 2013*; *Zeng et al., 2007*). The following UAS transgenic flies were used: UAS-DNrx (*Tian et al., 2013*; *Zeng et al., 2007*), UAS-SCAR-RNAi (THU 02179 .N; Center of Biomedical Analysis, Tsinghua University), UAS-DNlg1, UAS-DNlg1$^{SF-AA}$, UAS-DNlg1-EGFP, and UAS-DNlg1$^{SF-AA}$-EGFP. We generated UAS-DNlg1, UAS-DNlg1$^{SF-AA}$, UAS-DNlg1-EGFP, and UAS-DNlg1$^{SF-AA}$-EGFP by inserting the coding sequences into an attp-pUAST vector and injecting the resulting constructs into embryos of phiC31 attB-bearing flies (Bloomington stock center, #24870). The strategy used for the generation of EGFP-tagged DNlg1 was described previously (*Banovic et al., 2010*). We used the following Gal4 flies: motor neuron Gal4 OK6-Gal4 (Bloomington Stock center), muscle-specific Gal4: C57-Gal4 (*Budnik et al., 1996*), 24B-Gal4 (Kyoto Stock center), and Mef 2-Gal4 (Bloomington Stock center).

## NMJ staining and image analysis

The procedure for immunostaining the larval body-wall muscle was described previously (*Xing et al., 2014*). In brief, the body-wall muscles from third instar larvae were dissected in PBS solution and fixed them for 40 min with 4% paraformaldehyde. Then, the fixed samples were washed four times in 0.3% PBST (PBS + 0.3% Triton X-100), blocked them in blocking solution (1% BSA, 0.3% Triton X-100 in PBS) for 1 hr, and incubated them with primary antibody at 4°C overnight. The following primary antibodies were used in this study: mouse anti-α-Spectrin (DSHB, 1:25), mouse anti-DLG (DSHB, 1:50), mouse anti-SCAR (DSHB; 1:50), mouse anti-GluRIIA (DSHB; 1:50), rabbit anti-GluRIIB (1:1000) (*Tu et al., 2017*); rabbit anti-HRP (Jackson ImmunoResearch, 1:1000), and Texas Red®-X Phalloidin (Molecular probe, 1:6). AlexaFluor-488-conjugated, AlexaFluor-555-conjugated, and AlexaFluor-633-conjugated anti-mouse or anti-rabbit secondary antibodies (Invitrogen, 1:500) were used at room temperature for 1 hr. All images were collected using an LSM 710 Confocal Station (Zeiss) and processed with Adobe Illustrator CS6.

Image J software was used to quantify the relative F-actin areas and fluorescence intensities. For measurements of the relative F-actin areas, we selected single boutons and calculated the F-actin signal area and the anti-HRP signal area in each bouton using Image J. The relative F-actin area was expressed as the ratio of the F-actin area to the anti-HRP area. For measurements of fluorescence intensity, we set an arbitrary threshold for each channel based on the difference in intensity between the NMJ and the background regions. The sum of the pixels with intensities above the threshold was recorded by ImageJ. For comparison of fluorescence intensities between genotypes, all samples were processed simultaneously and under identical conditions. Similar to a previous study (*Zhao et al., 2015*), anti-HRP staining signal was used as a control signal. For F-actin staining, NMJ samples were incubated with a high concentration of Texas Red-conjugated phalloidin (1:6) for 13 min at room temperature. Images were collected at muscles 12/13, because the distance between the boutons and the myofibril in that muscle is relatively large, thus reducing the interference from a strong F-actin-stained background due to myofibril (*Ramachandran et al., 2009*). We performed statistical analyses as described in the source data tables.

Quantification of active zones was performed using a modified method according to Banovic's description (2010). The unopposed active zones were defined as active zones lacking GluR alignment. We co-applied GluRIIB rabbit polyclonal antibody and Brp mouse monoclonal antibody (nc82) for immunofluorescence staining. A branch of bouton clusters (usually 8–10 boutons) from muscle 4 of segment A3 were included in the analysis instead of quantifying only terminal boutons. To avoid subjective variation in the counting the numbers of non-overlapping Brp/GluRIIB spots, the ratio of non-overlapping Brp area to the total Brp area, namely percentage unopposed AZ area was used to represent the extent of unapposed active zones. That method is very similar to the way that the extent of co-localization of presynaptic and postsynaptic proteins in mammalian systems is quantified (*Singh et al., 2016*). The Brp spots in single-channel images and the overlapping spots of Brp and GluRIIB in corresponding two-channel merged images were respectively detected using ImageJ. Then, the total area of the Brp spots and that of the Brp/GluRIIB overlapping spots were quantified using the 'analyze particles' function in ImageJ. We defined the percentage apposed AZ area as the ratio of the total Brp/GluRIIB overlapping spot area to the total Brp spot area. The percentage unapposed AZ area was acquired by subtracting the percentage apposed AZ from 100%.

To quantify immunofluorescent signals on the cytoplasmic membrane or in the cytoplasm of S2 cells, plot profile analysis in ImageJ was performed according to the ImageJ user guide. The extent of SCAR recruitment from the cytoplasm to the cytoplasmic membrane was indicated by the ratio of the plasma membrane SCAR intensity (the mean value of two plot profile values on the membrane) to the cytoplasmic SCAR intensity (the mean value of all values within the cytoplasm).

All statistical analyses were performed using GraphPad Prism 7. We applied nonparametric one-way ANOVA followed by Tukey's post hoc test to evaluate the differences among multiple groups of samples. We used the Mann–Whitney U test to evaluate the difference between two groups of samples. The methods used for the statistical analyses in all statistical graphs are described in the source data.

## Immunoblotting analysis

Fly brain samples were homogenized with 1 × SDS loading buffer (50 mM Tris-HCl, 2% SDS, 0.1% bromophenol blue, 10% glycerol, 1% β-mercaptoethanol). The lysate samples and pull-down samples were separated by 10% or 8% SDS-PAGE and electro-transferred onto polyvinylidene difluoride membranes. We probed the immobilized proteins on the membranes with primary antibodies at 4°C overnight. The following primary antibodies were used in this study: mouse anti-SCAR (DSHB, 1:500) and mouse anti-Tubulin (Sigma, 1:10000). The samples were then incubated with HRP-conjugated secondary antibody at room temperature for 1.5 hr. The targeted proteins were visualized with SuperSignal West Pico Chemiluminescent substrate and SuperSignal West Femto Maximum Sensitivity substrate from Thermoscientific (Rockford, IL).

## GST pull-down assay

The procedure for the pull-down assay was described previously (*Nakao et al., 2008*). Briefly, GST protein and GST fusion proteins were expressed in BL21 cells with PGEX-5X-1 vector. We then purified and immobilized the cells with glutathione Sepharose 4B beads (GE Healthcare, SWE). WT fly heads were homogenized and lysed in lysis buffer (20 mM Tris-HCl, pH 7.5, 150 mM NaCl, 0.5% Triton X-100, 5% glycerol, 1 mM EDTA, 1 mM DTT, 1 mM PMSF, and 1 × Complete EDTA-free protease inhibitor cocktail (Roche, GER). Then, the lysate was incubated with GST protein or GST-tagged proteins immobilized on beads. We then washed the beads three times with lysis buffer and eluted them by boiling in 2 × SDS loading buffer.

## Electrophysiology

Third instar larvae were dissected and intracellular membrane potentials were recorded as previously described (*Jan and Jan, 1976*; *Sun et al., 2011*). Briefly, we chose muscle 6 in the A3 segment for recording. The free segmental nerve end was drawn into a microelectrode using an injector and stimulated with a Master-8 pulse stimulator (A.M.P.I, IL) at 0.3 Hz with a suprathreshold stimulating pulse and the electrodes (20–50 MΩ) filled with 3 M KCl for EJP recording. We recorded mEJPs for 20 s starting 8 s after EJP recordings. All recordings were conducted at room temperature with an Axoclamp 900A amplifier (Molecular Devices, Sunnyvale, CA) in bridge mode. The data were digitized with a Digitizer 1322A (Molecular Devices). We used Clampfit 10.2 to analyze the data.

## Cell culture, transfection, and staining

S2 cells were cultured on six-well cell culture plates overlaid with round glass cover slips (Corning) and incubated at 25°C in HyClone SFX-Insect Cell Culture Media (ThermoFisher Scientific, UK). We co-transfected the cells with PAC5.1/V5-His-A plasmid containing an actin 5C promoter and pUAST vector including the desired protein coding sequence using X-treme GENE HP (Roche) transfection reagent. Two days after transfection, the cells were fixed for 10 min with 4% formaldehyde (Sigma, GER), permeabilized with PBS solution containing 1% BSA and 0.1% TritonX-100, and incubated with primary antibodies (rabbit anti-GFP, 1:1000, Invitrogen; mouse anti-SCAR, 1:50, DSHB) at 4°C overnight. After washing the cells three times, we incubated them with Alexa-488-conjugated or Alexa-555-conjugated secondary antibodies at room temperature for 1 hr. We then washed the cells and mounted them in mounting media with DAPI (Vector Laboratories, Burlingame, CA). For siRNA assays, siRNAs were transfected with Lipofectamine 2000 (ThermoFisher Scientific, Grand Island, NY). After 3 days, the cells were collected for western blotting and immunostaining. The following siRNAs were used: Ctr-siRNA, ACGUGACACGUUCGGAGAATT; SCAR-siRNA, AUAGACAUUAAGCUUGUCGAG.

The S2 cell line was acquired from the China Center for Type Culture Collection (CCTCC). To test the mycoplasma contamination status, we used the MycoBlueTM Mycoplasma Detector kit (Vazyme, CHN), which works based on an improved short tandem repeat (STR) profiling technique. According to the kit manual, if the cell culture has mycoplasma contamination, the conserved sequence of mycoplasma DNA will be amplified by isothermal DNA polymerase, which will turn the reaction liquid from purple to blue. Thus, the results can be visualized by direct observation. All of the S2 cells used in this study were negative for mycoplasma contamination.

## Abbreviations

DNlg, *Drosophila* neuroligin; WRC, WAVE regulatory complex; NMJ, neuromuscular junction; GluR, glutamate receptor; GluRIIA, glutamate receptor IIA subunit; GluRIIB, glutamate receptor IIB subunit; EJP, excitatory junction potential; mEJP, miniature excitatory junction potential; SSR, subsynaptic reticulum; BRP, bruchpilot; DLG, Discs large 1; HRP, horseradish peroxidase; BMP, bone morphogenetic protein.

## Acknowledgements

We thank Dr Hermann Aberle for the *dnlg1* mutants, the Bloomington Stock Center for providing *Drosophila* stocks, DSHB for antibodies, and the members of the Xie laboratory for their critical comments on the manuscript and experimental design. This work was supported by the National Natural Science Foundation of China (31430035, 31171041 to WX and 31401107 to ML), the National Basic Research Program (973 Program) Grant (2012CB517903), NSFC-CIHR (81161120543 to WX and CCI117959 to ZPJ), and the China Postdoctoral Science Foundation (2015T80476 and 2014M560369 to ML).

## Additional information

### Funding

| Funder | Grant reference number | Author |
|---|---|---|
| National Natural Science Foundation of China | 31430035 | Wei Xie |
| National Natural Science Foundation of China | 31171041 | Wei Xie |
| National Basic Research Program | 2012CB517903 | Wei Xie |
| National Natural Science Foundation of China | 31401107 | Moyi Li |
| China Postdoctoral Science Foundation | 2015T80476 | Moyi Li |
| China Postdoctoral Science Foundation | 2014M560369 | Moyi Li |
| Canadian Institutes of Health Research | CCI117959 | Zhengping Jia |
| NSFC-CIHR | 81161120543 | Wei Xie |

The funders had no role in study design, data collection and interpretation, or the decision to submit the work for publication.

### Author contributions

Guanglin Xing, Data curation, Formal analysis, Validation, Investigation, Visualization, Methodology, Writing—original draft, Project administration, Writing—review and editing; Moyi Li, Data curation, Formal analysis, Supervision, Funding acquisition, Validation, Investigation, Methodology, Writing—original draft, Project administration, Writing—review and editing; Yichen Sun, Data curation, Formal analysis, Validation, Visualization; Menglong Rui, Data curation, Formal analysis; Yan Zhuang, Data curation; Huihui Lv, Formal analysis; Junhai Han, Methodology; Zhengping Jia, Conceptualization, Funding acquisition, Writing—review and editing; Wei Xie, Conceptualization, Formal analysis, Supervision, Funding acquisition, Investigation, Writing—original draft, Project administration, Writing—review and editing

### Author ORCIDs

Guanglin Xing http://orcid.org/0000-0002-8258-0293
Moyi Li http://orcid.org/0000-0003-3566-3931

Junhai Han 🔗 http://orcid.org/0000-0001-8941-2578
Zhengping Jia 🔗 http://orcid.org/0000-0003-4413-5364
Wei Xie 🔗 http://orcid.org/0000-0002-9179-4787

## Decision letter and Author response

Decision letter https://doi.org/10.7554/eLife.30457.028
Author response https://doi.org/10.7554/eLife.30457.029

## Additional files

**Supplementary files**
• Transparent reporting form
DOI: https://doi.org/10.7554/eLife.30457.026

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
