## [Decision Letter]

Thank you for submitting your article "Neurexin-Neuroligin 1 Regulates Synaptic Morphology and Function via WAVE Complex in *Drosophila* Neuromuscular Junction" for consideration by *eLife*. Your article has been evaluated by K VijayRaghavan (Senior Editor) and three reviewers, one of whom is a member of our Board of Reviewing Editors. The following individual involved in review of your submission has agreed to reveal his identity: Stephan J Sigrist (Reviewer #2).

The reviewers have discussed the reviews with one another and the Reviewing Editor has drafted this decision to help you prepare a revised submission.

Summary:

All three reviewers found that this manuscript addressed an important question in neuronal cell biology and synapse formation. We also agreed that the manuscript provided a series of compelling evidence that DNlg induce F-actin assembly in postsynaptic specializations. However, there are a number of technical questions that were raised by the reviewers. I have compiled a list of concerns/suggestions from all three reviewers. The "essential revisions" are raised by all three reviewers and the outcome of how you address these questions will significantly affect the outcome of the next round of reviews.

Essential revisions:

1) Quantify the apposition between active zones and GLuRs in perspective mutants ("well resolved immunofluorescence stainings versus glutamate receptors and active zone components such as BRP, followed by quantification of number of active zones/PSDs and (lack of) apposition frequency". This analysis could essentially follow Figure 4. See Banovic et al., Neuron 2010 (DOI 10.1016/j.neuron.2010.05.020).

2) Analyse whether Neuroligin 2 mutant also leads to loss of the WRC, and if a Neuroligin 1/Neuroligin 2 double mutant gives the same phenotype as the single mutants.

3) Add quantification to the following figures: Figure 3, Figure 6, Figure 1—figure supplement 1, Figure 4—figure supplement 1, Figure 5—figure supplement 1, and Figure 6—figure supplement 1. At least the main figures need to be quantified.

[Editors' note: further revisions were requested prior to acceptance, as described below.]

Thank you for submitting your article "Neurexin-Neuroligin 1 Regulates Synaptic Morphology and Function via WAVE Complex in *Drosophila* Neuromuscular Junction" for consideration by *eLife*. Your article has been reviewed by two peer reviewers, and the evaluation has been overseen by a Reviewing Editor and K VijayRaghavan as the Senior Editor. The following individual involved in review of your submission has agreed to reveal his identity: Stephan J Sigrist (Reviewer #2).

The reviewers have discussed the reviews with one another and the Reviewing Editor has drafted this decision to help you prepare a revised submission.

The revised manuscript was reviewed by the same reviewers. I am glad to report that the reviewers felt that the manuscript is significantly improved and they are now recommending publication given that you can address several remaining points. The most significant point was point No. 1, where you misunderstood the reviewers request in the first place. Please address these points and please find a professional editing service to do language editing. Thank you.

1) The data that need clarification are related to Figure 3, about membrane recruitment of the endogenous WRC by overexpression of DNlg1-EGFP. In the review question 4): The S2 cell assay in Figure 3 suggests that nlg binding to SCAR is direct. From structural analysis, WIRS binds to the interface between two WRC components. It is surprising that this recruitment works so well, considering the possible low abundance of the endogenous WRC. Suggestion: one of these two approaches can help to establish this pointa) RNAi or KO of SCAR would be good to validate if the observed change is actually due to WAVE, instead of something nonspecific.b) Alternatively, they can do F-actin staining.

The authors apparently misunderstood the suggestions, which were to validate the recruitment by RNAi or KO of SCAR, or F-actin staining in S2 cells. Although as their response to this question, they added experiments of SCAR RNAi in *Drosophila*, which gave good results, the membrane recruitment assay in S2 cells remains the same and is not convincing, especially now with the new quantification line scans and bar graph. Here are my questions:a) It is much easier to obtain a large number of S2 cells for quantitative imaging than animals. The current p-value is only <0.05, which could be significantly improved by having larger sample size. Why did the authors only use 10 cells for each sample for statistical analysis? How were the 10 cells selected? What are the procedures to avoid subjective bias when selecting the 10 cells?

b) Compare Figure 3 to J and L panels, the membrane recruitment of SCAR signal was dramatic with WT DNlg1 expression. In Figure 3K4, the contrast of membrane SCAR vs. cytoplasmic SCAR is ~40-60 fold. However, in Figure 3, the fold change of SCAR at membranes is only 1.2. If this is my misunderstanding, please make the figure labels more self-explanatory. If this is because the images used in panel J, K and L are not true reflection of the 1.2 fold change, the authors should use more representative images, not the most dramatic images, to avoid misleading the audience and potential users of this method.

2) I cannot see a real difference comparing the Scar signal in *dnlg1* and *dnlg2* mutants. Could they maybe show like a higher magnification here?

3) Finally, it remains surprising that lack of DNlg2 (though this protein lacks the binding motif) provokes a similar F-actin deficit as does absence of DNlg1. Given the amount of data they already provide, a deeper analysis of this relation can await future analysis. Can you at least provide a more detailed discussion on this point?

---

## [Author Response]

Essential revisions:1) Quantify the apposition between active zones and GLuRs in perspective mutants ("well resolved immunofluorescence stainings versus glutamate receptors and active zone components such as BRP, followed by quantification of number of active zones/PSDs and (lack of) apposition frequency". This analysis could essentially follow Figure 4. See Banovic et al., Neuron 2010 (DOI 10.1016/j.neuron.2010.05.020).

As suggested, we quantified the apposition between active zones (BRP) and GluRIIB in perspective mutants (Figure 6). Differing from the previous study of Banovic et al., we used a branch of bouton clusters instead of terminal boutons for statistical analysis. To quantify misalignment between BRP spots and GluRIIB, we calculated the ratio of unapposed area to the total area of all BRP spots in a chain with multiple boutons. This is very similar with the method applied in mammalian system to quantify colocalization extent of presynaptic and postsynaptic proteins (Singh et al., 2016). Instead of numbering unapposed BRP spots, we chose to calculate the unopposed area percentage as it’s sometimes hard to distinguish if BRP and GluRIIB signals are partially overlapped or they are really unopposed. We added a paragraph in Materials and methods section to describe our method in detail too.

You might notice that the percentage numbers we observed are different to the numbers reported in Banovic’s work (WT: 1.4% unopposed AZ; *dnlg1* mutant: 15.7% unopposed AZ). That is because we used different index (area percentage rather than number percentage) to reflect this phenotype. But we did detect similar difference between WT and *dnlg1* mutants, which is close to 15-20% between WT and *dnlg1*, and we found rescue-AA could not completely rescue this phenotype as rescue-WT (Figure 6). Meanwhile, we quantified type Ib boutons completely losing GluR receptors (orphan boutons) in perspective mutants (Figure 6). Similar to what was reported in Banovic’s article, we observed ~10% of orphan bouton in *dnlg1* mutants. But rescue-AA and rescue-WT have similar frequency of orphan bouton as WT. These data suggest that F-actin reduction might be at least the partial reason for GluR mislocalization in *dnlg1* mutants.

2) Analyse whether Neuroligin 2 mutant also leads to loss of the WRC, and if a Neuroligin 1/Neuroligin 2 double mutant gives the same phenotype as the single mutants.

As suggested, we analyzed the SCAR distribution in neuroligin 2 mutant (Figure 3), and found that loss of DNlg2 also caused a remarkable decrease in SCAR level. We also examined postsynaptic F-actin and α-Spectrin in *dnlg1;dnlg2* double mutants and found that the double showed similar phenotypes to that of *dnlg1* single mutants (Figure 1). We discussed the relationship between DNlg1 and DNlg2 in regulation of postsynaptic F-actin assembly in Discussion section.

3) Add quantification to the following figures: Figure 3, Figure 6, Figure 1—figure supplement 1, Figure 4—figure supplement 1, Figure 5—figure supplement 1, and Figure 6—figure supplement 1. At least the main figures need to be quantified.

As suggested, we added quantitative analysis to the Figure 3I-K (new Figure 3), Figure 6 (new Figure 7,Figure 8), Figure 1—figure supplement 1 (new Figure 1—figure supplement 1), and Figure 5—figure supplement 1 (new Figure 6—figure supplement 1). We combined the upper panels of Figure 6A-C with Figure 6—figure supplement 1 into the new Figure 7 and added quantitative analysis to this new Figure 7. The lower panels of former Figure 6D-F and related quantification analysis formed the new Figure 8 . Only Figure 4—figure supplement 1 (new Figure 5—figure supplement 1) was not quantified, because this data was used to address that DNlg1^SF-AA^ could target to postsynaptic site as WT DNlg1.

[Editors' note: further revisions were requested prior to acceptance, as described below.]

The revised manuscript was reviewed by the same reviewers. I am glad to report that the reviewers felt that the manuscript is significantly improved and they are now recommending publication given that you can address several remaining points. The most significant point was point No. 1, where you misunderstood the reviewers request in the first place. Please address these points and please find a professional editing service to do language editing. Thank you.1) The data that need clarification are related to Figure 3J-L , about membrane recruitment of the endogenous WRC by overexpression of DNlg1-EGFP. In the review question 4): The S2 cell assay in Figure 3J-L suggests that nlg binding to SCAR is direct. From structural analysis, WIRS binds to the interface between two WRC components. It is surprising that this recruitment works so well, considering the possible low abundance of the endogenous WRC. Suggestion: one of these two approaches can help to establish this pointa) RNAi or KO of SCAR would be good to validate if the observed change is actually due to WAVE, instead of something nonspecific.b) Alternatively, they can do F-actin staining.The authors apparently misunderstood the suggestions, which were to validate the recruitment by RNAi or KO of SCAR, or F-actin staining in S2 cells. Although as their response to this question, they added experiments of SCAR RNAi in Drosophila, which gave good results, the membrane recruitment assay in S2 cells remains the same and is not convincing, especially now with the new quantification line scans and bar graph. Here are my questions:a) It is much easier to obtain a large number of S2 cells for quantitative imaging than animals. The current p-value is only <0.05, which could be significantly improved by having larger sample size. Why did the authors only use 10 cells for each sample for statistical analysis? How were the 10 cells selected? What are the procedures to avoid subjective bias when selecting the 10 cells?

We are sorry for misunderstanding reviewer’s suggestions. As suggested, we performed RNA interference with SCAR siRNA in S2 cells this time. Compared with untreated S2 cells or control siRNA treated cells, SCAR protein level in SCAR siRNA treated S2 cells was dramatically reduced. This thus demonstrated specificity of immunostaining signals from recognition of SCAR protein by its antibody in this cell line. And it is clearly proved that the observed changes of SCAR distribution in DNlg1-EGFP or DNlg1^SF-AA^-EGFP overexpressed S2 cells are actually due to WAVE, instead of something nonspecific. Results of RNAi in S2 cells are presented in new Figure 3—figure supplement 2.

In our previous version of Figure 3, we chose images of the first 10 cells scanned in each group for quantification because this N number was already enough to show us statistical significance between tested groups. And we thought if we always chose the first 10 cells microscopically scanned for each group, this was a way to avoid subjective bias. But we absolutely agree with reviewers that what we used is not an ideal N number for statistical analysis for S2 cells especially considering our p value (only <0.05) and easy availability of these cells. And the way we avoid subjective bias is not very strict too. As suggested, in this revised version, we increased N number to around 25 cells. And our p value reached to <0.001(new Figure 3 and Figure 3—source data 1) this time.

b) Compare Figure 3 to J and L panels, the membrane recruitment of SCAR signal was dramatic with WT DNlg1 expression. In Figure 3K4, the contrast of membrane SCAR vs. cytoplasmic SCAR is ~40-60 fold. However, in Figure 3, the fold change of SCAR at membranes is only 1.2. If this is my misunderstanding, please make the figure labels more self-explanatory. If this is because the images used in panel J, K and L are not true reflection of the 1.2 fold change, the authors should use more representative images, not the most dramatic images, to avoid misleading the audience and potential users of this method.

Thank you very much for pointing out our problem with not using the most representative image of DNlg1-EGFP but instead using a more dramatic image of this group in our old Figure 3. We realized that this could be misleading to potential users of this method too. As suggested, in this revised version, we changed Figure 3 to more representative images (new Figure3K1-K3). Furthermore, we improved our quantification strategy for SCAR intensity plot profile analysis. In our previous analysis, the extent of SCAR recruitment from cytoplasm to cytoplasmic membrane was indicated using the ratio of plasma membrane SCAR intensity (the mean value of two plot profile values on the membrane) to nearby cytoplasmic SCAR intensity (the mean value of two neighbor-peak values in the cytoplasm). Now we turn to using the ratio of plasma membrane SCAR intensity (the mean value of two plot profile values on the membrane) to the cytoplasmic SCAR intensity (the mean value of all values within cytoplasm) to indicate this event. And we do get a more significant difference between DNlg1-EGFP and DNlg1^SF-AA^-EGFP using this new quantification strategy and increasing N number (new Figure 3). We thought all these changes will further improve the stringency of our data and make our conclusions more convincing.

2) I cannot see a real difference comparing the Scar signal in dnlg1 and dnlg2 mutants. Could they maybe show like a higher magnification here?

As suggested, we added the high-magnification views of selected regions (new Figure 3D3, E3, F3, G3 and H3 blue dash line rectangles) at the right side of the original images. It showed that SCAR proteins in *dnlg1* mutants don’t tightly surround synaptic boutons (Figure 3G1 and G3 high-magnification images, yellow arrow heads in Figure 3G1 and G3 indicated diffused SCAR signals) compared with WT controls and *dnlg2* mutants.

3) Finally, it remains surprising that lack of DNlg2 (though this protein lacks the binding motif) provokes a similar F-actin deficit as does absence of DNlg1. Given the amount of data they already provide, a deeper analysis of this relation can await future analysis. Can you at least provide a more detailed discussion on this point?

As suggested, we added more detailed discussion on how DNlg2 possibly regulates postsynaptic F-actin in Discussion section. Briefly, the first possibility is that DNlg1, DNlg2, DNrx, and the WRC may form a large protein complex, so that single deletions of any of the components of the complex may have similar effects on actin regulation. The second possibility is that DNlg2 regulates F-actin via p21 protein (Cdc42/Rac)-activated kinase (DPAK), an important regulator of F-actin assembly that has been shown to be downregulated in *dnlg2* mutants (Sun et al., 2011). Another possibility is that as the PDZ-binding motif of mouse Nlg1 can bind to spine-associated Rap guanosine triphosphatase–activating protein (SPAR) and subsequently activate the LIMK1/cofilin pathway to regulate actin filaments, spine morphology, and plasticity (Liu et al., 2016), it is reasonable to explore whether the PDZ-binding motif of DNlg2 can bind to Rap GTPase activating protein 1 (RapGAP1), the *Drosophila* homolog of SPAR.